



*D*ynamic *A*nthropogenic activitie*S* impacting *H*eat emissions (DASH v1.0): Development and evaluation

Isabella Capel-Timms[1,2], Stefán Thor Smith[2], Ting Sun[1], Sue Grimmond[1]

[1] Department of Meteorology, University of Reading, UK  RG6 6BB

[2] School of Built Environment, University of Reading, UK RG6 6DF

Corresponding Author: c.s.grimmond@reading.ac.uk

**Abstract**

Thermal emissions or anthropogenic heat fluxes ($Q_F$) from human activities impact the local and larger scales urban climate. DASH considers both urban form and function in simulating $Q_F$ by use of an agent-based structure that includes behavioural characteristics of city populations. This allows social practices to drive the calculation of $Q_F$ as occupants move, varying by day type, demographic, location, activity, socio-economic factors and in response to environmental conditions. The spatial

resolution depends on data availability. DASH has simple transport and building energy models to allow simulation of dynamic vehicle use, occupancy and heating/cooling demand, with subsequent release of energy to the outdoor environment through the building fabric. Building stock variations are captured using archetypes. Evaluation of DASH in Greater London for various periods in 2015 uses a top-down inventory model (GQF) and national energy consumption statistics. DASH reproduces the expected spatial and temporal patterns of $Q_F$ but the annual average is smaller than published energy data.

Overall the model generally performs well, including for domestic appliance energy use against top down model results. DASH could be coupled to an urban land surface model and/or used offline for developing coefficients for simpler/faster models.

**Keywords:** Anthropogenic heat emission; dynamic model; energy, urban climate, London

**Notation** (with location of 1$^{st}$ mention)

|  | Description | 1$^{st}$ |
|---|---|---|
| $\alpha, \alpha_{j,A_N}, \alpha_{j,k}, \alpha_{j,u}$ | Characteristic of appliance $\alpha$ of type $j$: quantity in $A_N$, domestic usage factor $u$, market permeation $k$ | 2.4.3 |
| $a_D^N, a_E^N, a_H^N, a_O^N, a_R^N, a_W^N$ | Domestic, primary school, secondary school, other (e.g. leisure) shop and work subareas | 2.1 |
| $a_D^{senior}, a_D^{working}, a_D^{young}$ | Dominant age cohort characteristics of subareas (analysed): seniors, working adults and young people (infants, children or teenagers) | 4 |
| $a_S^N$ | Subarea of $A_N$ with specific activity $s$ occurring | 2.1 |
| $A$ | Building surface area (m$^2$) | B |
| AADT | Annual Average Daily Traffic | 3.1 |
| ABM | Agent-based model | 2 |
| $AE_i$ | Absolute error ($|\Delta_i|$) | 3.2 |
| $A_N$ | Spatially discrete agent | 2 |
| $AnE_i$ | Absolute normalised error | 3.2 |
| $AO$ | Consumption class: active only | 2.4.3 |
| API | Application programming interface | 2.4.2 |
| $AS$ | Consumption class: active/standby | 2.4.3 |
| $\beta$ | Bowen ratio ($Q_H/Q_E$) | 2.4.1 |
| $B$ | Spatial unit, may be coarser than $A_N$ | 2.2 |



| | | |
|---|---|---|
| $C$ | Consumption class: continuous | 2.4.3 |
| CBD | Central business district | 6 |
| $C_{m,r}$ | Mode-appropriate ratio for $m$ on $r$ ($n_{o,m,r}$ vehicle$^{-1}$) | 2.4.2 |
| $c, c_p$ | Specific heat capacity, specific heat capacity of air at constant pressure (J kg$^{-1}$ K$^{-1}$) | 2.4.3 |
| $\Delta_i$ | Model-observation (reference) difference for variable $i$ | 3.2 |
| DASH | *Dynamic Anthropogenic activitieS impacting Heat emissions*) | 1 |
| DHW | Domestic hot water | 2.4.3 |
| $d_{i,j}$ | Distance between origin $i$ and destination $j$ (m) | 2.2 |
| $\varepsilon$ | Emissivity | B |
| $f$ | Fuel type | 2.4.2 |
| $f_{x,\alpha_j}$ | Fraction of households with $x$ active occupants using $\alpha_j$ | 2.4.3 |
| $F_{m,f}$ | Heat emission with fuel type $f$ for $m$ (W m$^{-1}$) | 2.4.2 |
| $\Gamma_{i,j}$ | Gravity weighting for all potential trips between origin $i$ and destination $j$ | 2.2 |
| GIS | Geographical information system | 2.4.2 |
| GL | Greater London | 3.1 |
| GQF | GQF model (Gabey et al., 2019) | 3.2 |
| $h$ | Convection coefficient (W m$^{-2}$ K$^{-1}$) | 3.1 |
| $HC$ | Heating and cooling usage | 2.4.3 |
| $HW$ | Hot water usage | 2.4.3 |
| IQR | Interquartile range | 3.2 |
| $\kappa$ | System efficiency | 2.4.3 |
| $K_\downarrow$ | Downwelling shortwave radiation (W m$^{-2}$) | 2.4.3 |
| $k_e$ | Effective thermal conductivity (W m$^{-1}$ K$^{-1}$) | 3.1 |
| $l$ | Lighting | 2.4.3 |
| $l_{base}$ ($l_{min}/l_{max}$) | Base (min/max) luminous intensity | 2.4.3 |
| $L$ | Thickness of building component (m) | B |
| LA | Local Authority | 3.1 |
| $L_m$ | Length of unit vehicle for $m$ (m) | 2.4.2 |
| LOWESS | Locally Weighted Scatterplot Smoothing | 2.4.2 |
| $L_{r,t}$ | Distance travelled in $t$ (m) | 2.4.2 |
| LSOA | Lower-level Super Output Area | 3.1 |
| $m$ | Travel mode (e.g. car, bus, train, walk) | 2.4.2 |
| $M$ | Metabolic rate (W) | 2.4.1 |
| MSOA | Mid-level Super Output Area | 3.1 |
| $n_b$ ($n_{b,x}$) | Number of households (with $x$ active occupants) | 2.4.3 |
| $n_{o,m,r}$ | Number of occupants for $m$ on $r$ | 2.4.2 |
| $nE_i$ | Normalised error | 3.2 |
| $nMax$ | Maximum-normalised value | 3.2 |
| NS | Non-school weekday | 4 |
| OA | Output area | 3.1 |
| $O_C$ | Occupant | 2 |
| $\pi(t)$ | Stationary distribution for state at time step $t$ | A |
| $\Psi_{b/g/s/i}$ | View factor for buildings/ground/sky/surface $i$ | B |
| $P, P_{max}$ | Power rating, maximum power rating (W) | 2.4.3 |
| $p(t)_{m,n}$ | Transition probability from state $m$ to state $n$ at time step $t$ | A |
| $q, q_H, q_C$ | Energy use (for heating, cooling) (W) | 2.4.3 |
| $q_{cd}$ | Building conductive flux (W) | B |



| | | |
|---|---|---|
| $q_{cv}$ | Building convective flux (W) | B |
| $Q^*$ | Net all-wave radiation (W m$^{-2}$) | 1 |
| $Q_E$ | Turbulent latent flux (W m$^{-2}$) | 1 |
| $Q_{F,(B/M/T)}$ | Anthropogenic heat flux (emissions from buildings/metabolic activity/transport) (W m$^{-2}$) | 1 |
| $Q_{F,B}^\alpha, Q_{F,B}^{HC},$ $Q_{F,B}^{HW}, Q_{F,B}^l$ | $Q_{F,B}$ from: appliance usage, heating and cooling, hot water demand, lighting (W m$^{-2}$) | 2.4.3 |
| $Q_{F,B}^{elec}, Q_{F,B}^{gas}$ | $Q_{F,B}$ from: electricity, gas consumption | 5 |
| $Q_H$ | Turbulent sensible flux (W m$^{-2}$) | 1 |
| $Q_{L*}, Q_{L\uparrow}$ | Net longwave radiation. Outgoing longwave radiation (W m$^{-2}$) | B |
| $\Delta Q_S$ | Net storage heat flux (W m$^{-2}$) | 1 |
| $q_{vent}$ | Energy loss/gain from ventilation (W) | 2.4.3 |
| $\rho\ (\rho_a)$ | Density (of air) (kg m$^{-3}$) | 2.4.3 |
| $r$ | Route type $r$ (e.g. minor- or major-road, over-ground- or below-ground-rail) | 2.4.2 |
| $R_{lim}$ | Route capacity limit | 2.4.2 |
| $\sigma$ | Stefan-Boltzmann constant (W m$^{-2}$ K$^{-4}$) | B |
| STEBBS | **S**implified **T**hermal **E**nergy **B**alance for **B**uilding **S**cheme | 2.4.3 |
| SW | School/work-day | 4 |
| $T$ | Time step (e.g. ten minutes) | 2.3 |
| $\tau$ | Effective transmissivity | B |
| $\Theta$ | Albedo | B |
| $t_b$ | Journey specific time bin | 2.4.2 |
| $T_{f/s/si/so}$ | Temperature or fluid $f$/surface $s$/indoor surface $si$/outdoor surface $so$ (K) | B |
| $T_i$ | Internal water/air temperature (K) | 2.4.3 |
| $T_o$ | Outdoor air temperature (K) | 2.4.3 |
| ToU | Time of use | 3.1 |
| $T_{set}$ | Setpoint temperature (K) | 2.4.3 |
| TUS | Time Use Survey | 3.1 |
| UK | United Kingdom | 3.1 |
| $V_{FR}, V_R$ | Volumetric flow rate, ventilation rate (m$^3$ s$^{-1}$) | 2.4.3 |
| $V_{m,r}$ | Number of unit vehicles for $m$ on $r$ | 2.4.2 |
| $v, v_r, v_{r,lim}$ | Speed, speed of travelling vehicle on $r$, speed limit on $r$ (m s$^{-1}$) | 2.4.2 |
| $V_T$ | Volume of water tank (m$^3$) | 3.1 |
| WD | Weekday | 3.2 |
| $ws$ | Wind speed (m s$^{-1}$) | 3.1 |
| WWR | Window-to-wall ratio | 3.1 |
| $X_i\ (X_{M,i}, X_{O,i})$ | Output (M: modelled, O: observed/reference) value | 3.2 |

# 1 Introduction

The anthropogenic heat flux, $Q_F$, the thermal emissions arising from metabolic, chemical and electrical energy use, is an

additional energy source in the urban surface energy balance. As $Q_F$ is a function of human activity that can be associated with a range of spatial and temporal scales, it impacts the local-scale weather and climate in cities. For example, heating of buildings in cold climates can be an important influence on the urban heat island (UHI) (Hinkel et al., 2003; Bohnenstengel et al., 2014), whilst in summer the additional heat release from air conditioning (De Munck et al. 2013; Salamanca et al. 2014) can elevate air temperatures. The impacts of additional heat may exacerbate heat-related mortality rates during

heatwaves in urban areas (Heaviside et al., 2016) and increase electricity consumption in warmer weather (Santamouris et



al., 2001). Although there are multiple methods to estimate anthropogenic heat emissions, it has often been ignored in urban climate studies (Sailor, 2011).

The feedback of $Q_F$ on the other surface energy balance fluxes can be important (Bueno et al., 2012; Best and Grimmond,
2016; Ward et al., 2016). The surface energy balance is a fundamental driver of the atmospheric processes within the urban boundary layer, across a range of spatial and temporal scales (Oke, 1988):

$$Q^* + Q_F = Q_H + Q_E + \Delta Q_S \qquad (\text{W m}^{-2}) \qquad (1)$$

where $Q^*$ is the net all-wave radiation, $Q_F$ the anthropogenic heat flux, $\Delta Q_S$ the net storage heat flux, $Q_H$ the turbulent sensible and $Q_E$ turbulent latent heat fluxes. These fluxes are partly responsible for the energy involved in the transfer of
heat, mass and momentum (Oke, 198) and the stability of the urban boundary layer. The three major source terms of $Q_F$ (Grimmond, 1992):

$$Q_F = Q_{F,B} + Q_{F,M} + Q_{F,T} \qquad (\text{W m}^{-2}) \qquad (2)$$

relate to buildings ($Q_{F,B}$), metabolic (people, animals) activity ($Q_{F,M}$), and transport ($Q_{F,T}$). As a result, the daily movement of people through a city will have a local, short term effect, whilst the widespread uptake of new technologies (e.g. energy
efficient appliances) could have a city-wide, long term consequence.

There are multiple approaches to estimate $Q_F$ (Sailor, 2011). Using population data, top-down methods disaggregate energy consumption and traffic data to produce diurnal profiles of $Q_F$ (Sailor and Lu, 2004; Lee et al., 2009; Allen et al., 2011; Ferreira et al., 2011; Iamarino et al., 2012; Lindberg et al., 2013; Lu et al., 2015) Although constrained by data availability,
such approaches can be updated quickly to provide representative values of past states for large areas (Gabey et al., 2019). However, there is little variation between days, as the models tend to use static diurnal profiles. For example, the flow of people between residential and work areas does not respond to potential events that cause actual changes (e.g. blocked roads from flooding) and is assumed to be homogeneous across a city (Iamarino et al. 2012). Energy is often assumed to be released directly to the outdoor environment (Sailor, 2011) rather than indoors. Whilst aggregate behaviour may be captured,
the heterogeneity in processes (e.g. appliance use, technology uptake, changing work practices) are missed despite components (of eq. 2) being determined. Top-down approaches do though provide a basis to assess other approaches as their aggregate output is based on metered data.

Bottom-up models exist for the different types of heat emissions (of eq. 2) from buildings (e.g. Kikegawa et al. 2003; Bueno
et al. 2012; Schoetter et al. 2017), transport (e.g. Smith et al. 2009), and metabolism (e.g. Thorsson et al. 2014). Individually, they provide some information about behavioural and system change impacts on energy use and heat emissions. For example, building heat releases to the outdoor environment can be modified by building design (e.g. material conduction) and occupancy behaviours (e.g. ventilation, heating systems); and metabolic models capture activity and metabolic types (e.g. adults, children, animals). Other methods to estimate $Q_F$ include assuming energy balance closure (Offerle et al., 2005;
Pigeon et al., 2007; Crawford et al., 2017; Chrysoulakis et al., 2018) in eq. 1 with all other terms measured or estimated, and measurements of component fluxes (e.g. Kotthaus and Grimmond 2012).

Whilst existing models of $Q_F$ give plausible estimates, they typically do not capture changes resulting from human behaviour in small areas as city-wide assumptions are used when finer spatial resolutions are unavailable. Thus, not all techniques can
identify $Q_F$ hotspots (Gabey et al., 2019). Moreover, the processes causing change in anthropogenic energy use need to be modelled so the dynamic nature of $Q_F$ and implications of disruption to social practices to $Q_F$ can be investigated. Capturing the interplay between energy related behaviours and meteorological conditions should help exploration of system feedbacks and resulting effects on urban climates and city activities.



The terms of eq. 2 vary with land use and activity within an area creating spatial and temporal heterogeneity of $Q_F$. In turn, this impacts the urban surface energy balance (eq. 1). Models that can respond to influencing factors allow changes to be understood and potentially managed. Changes may occur at different spatial and temporal scales, for example: (i) city-wide building stock (e.g. type, dimension, materials) changes at decadal time-scales impact heating and cooling needs (i.e. modifying $Q_{F,B}$); (ii) individuals numerous sub-daily activity and travel decisions impact all three components at the

microscale; (iii) social-cultural practices play out across large spatial and temporal extents; (iv) transport dynamics can be modified over small spatiotemporal scales (e.g. road closures) or large spatial and temporal extents through technology (e.g. fuel, transport) and policy/planning (e.g. speed limits in neighbourhoods) changes.

Human behaviour and regional climate can impact each source term of $Q_F$. High- to mid-latitude cities with colder climates

use winter space heating, whereas in hotter climates air-conditioning in summer (Sailor and Lu, 2004) is increasingly used. Work schedules and other culturally informed practices (e.g. social eating, religious worship) alter the time of day, day of week, and time of year (i.e. national holidays) that energy demand occurs (Allen et al. 2011). These influences are not addressed by many static models (Allen et al. 2011, Dong et al., 2017) and associated dynamics are neglected despite impacting emissions spatially (e.g. Björkegren and Grimmond, 2018).


Here we present a new bottom-up model for $Q_F$ (DASH, ***D**ynamic **A**nthropogenic activitie**S** impacting **H**eat emissions*) that captures city features (i.e. place), people's activities, demographics, variations in building-type (e.g. thermal properties), and transport energy use and heat release. The model allows impacts from activities and interactions across a wide range of spatial and temporal scales to be explored by taking an agent-based approach. With both the heterogeneity of city energy use

and dynamics of the whole city captured by DASH, comparisons to top-down inventories or other data with coarser spatial and temporal scale resolutions are possible. These patterns can be analysed to diagnose the sensitivity of the steady-state to events that cause perturbations in agent-level behaviour. The general model structure and functionality are described (section 2). It is applied (section 3) and evaluated (section 4) in Greater London using inventory based results (Gabey et al., 2019).

## 2     Model development

As DASH takes an agent-based approach, all processes have either an interaction or reaction of agents (Macal and North, 2010). The agents represent the decisions for movement and citizens activities (e.g. cooking) that impact energy use and therefore $Q_F$. The dynamics result from agent activity in multiple processes that exist in each $Q_F$ source term (Fig. 1a) but share outputs (Fig. 1b). For each spatially scalable agent (section 2.1) there is (Fig. 1a):

1) *An agent-based model (ABM) scheduler*: to capture the evolutionary dynamics (section 2.2) of the spatially-discrete

110        agents $A_N$.

2) *Three source-specific $Q_F$ estimators*: use movement and activity from the ABM scheduler to model metabolic ($Q_{F,M}$, section 2.4.1) and transport-related ($Q_{F,T}$, section 2.4.2) anthropogenic heat. Given the dominant role of building energy use to urban anthropogenic heat (Sailor and Lu, 2004; Pigeon et al., 2007; Allen et al., 2011; Sailor, 2011; Nie et al., 2014; Zheng and Weng, 2017; Gabey et al., 2019), a building energy model (section 2.4.3 and Appendix B) is

115        integrated within DASH to estimate $Q_{F,B}$ by accounting for occupant behaviour that impacts both appliance energy use and indoor environmental conditioning.

The main DASH workflow is driven by agent-agent interactions with a three-stage process determining $Q_F$ per time step (Figure 1b):

*Stage 1:* Agent-agent interaction occurs through occupant ($O_C$) exchange processes (blue, Fig. 1b) that are modified by demographics as well as type and time of day.





*Stage 2:* Occupancy levels associated with an agent (yellow, Fig. 1b) modify appliance energy use ($P_\alpha$, Fig. 1), building heating and cooling control (via the building energy model, STEBBS), and volume of vehicles on the transport network (green, Fig. 1).

*Stage 3:* Source-specific $Q_{F,B}$, $Q_{F,T}$ and $Q_{F,M}$ terms are calculated for each agent and combined to give $Q_F$ for each agent's geographical region.

All processes operate at the same spatial unit (rather than area) and time step. These are both defined by the data used to inform the ABM scheduler. Rules that govern the processes may be informed by data and actions at coarser scales.

**2.1    Spatial granularity**

In agent based model design there is flexibility as to what "agents" represent; for example, individuals, households, spatial areas, or businesses (Crooks and Heppenstall, 2012; O'Sullivan et al., 2012). However, the chosen units should be able to interact with each other and respond. The constraints on selecting the most suitable entity for an agent include the aim, data availability and computer resources. In DASH, agents represent spatial units that interact by exchange of occupants - the

number, activity and type of which informs the calculations of $Q_F$ (Fig. 1).

A spatial unit's $Q_F$ depends on the number and composition of occupants' characteristics and their activities. For example, residential areas experience an increase in $Q_{F,B}$ as occupants wake up and start to use appliances or heating/cooling. As they leave home, $Q_{F,T}$ increases as fuel is used for transport and as the $O_C$ are passed between agents the changing activity and

occupancy numbers impact on each agent's $Q_F$. By using spatial units as agents (with $O_C$ as an agent property), agents can be scaled according to behavioural data and computational constraints. The relationship of agents to occupants can be from many-to-one and many-to-many. Here a many-to-many relationship is used given the computational and data constraints.

The agents interact by exchanging $O_C$ based on rules associated with the number, type, and activities of occupants. These are

also used in calculation of the energy use of an agent, i.e. the agents' response. Agent representation is designed to be data-driven (analysed) and so behaviour is constrained by data availability. For individual cities, the context (social, physical) provides the agents probable ('exact') characteristics, while administrative boundaries from national census (or other large survey data) will typically constrain DASH.

The agent ($A_N$) based spatial unit (as determined by data availability) contains subareas ($a_S^N$) of activity (not spatial units) to which the $O_C$ are assigned. Hence, population statistics are needed to characterise subareas. The subarea notation identifies the agent (superscript) and activity area (subscript). In this version of the model, there are six subareas: (i) domestic ($a_D^N$), (ii) workplace ($a_W^N$), (iii) primary school ($a_E^N$), (iv) secondary school ($a_H^N$), (v) shop ($a_R^N$), and/or (vi) other ($a_O^N$). There is a minimum of one subarea in each $A_N$, with the total number and type in each $A_N$ to be determined according to available data

and city context (e.g. a commercial district may only consist of $a_W^N$). Despite the $A_N$ location being static their properties are dynamic.

As $A_N$ have the decision-making capability for exchanging $O_C$, they interact by 'releasing' or 'accepting' occupants. Spatial variation in $O_C$ exchange is provided by the characteristics of the $a_S^N$, for example $a_W^N$ with higher workday populations being more likely to accept occupants during workday hours than other $a_W^N$ with smaller workday populations. Temporal

variability is governed by aspects of human behaviour, with granularity provided by different categories of $O_C$ identified within the data used to inform *the ABM scheduler*. The model can, therefore, capture differences associated with time of day, day of week, type of day (e.g. holiday or not) and time of year within (and across) different $O_C$ categories. Thus, this design results in the spatiotemporal dynamics of $Q_F$.




Each $A_N$ is located within larger spatial units ($B$) to allow coarser resolution spatial data to inform model behaviour (e.g. traffic speed limits, school districts), as well as enabling different spatial representation of $Q_F$ in analysis. Note that there can be multiple levels of directly nested spatial units. This permits different level of data availability and governance structure (e.g. impacting decision making/options) to be appropriately captured. Hence, impacts from changes in small areas on the

surroundings can be explored.

### 2.2     Rules of $A_N$ interaction

$O_C$ are generated and assigned to categories used to inform energy demand behaviour and movement (e.g. age, work). To enable movement of $O_C$, they are each associated with subarea types $a_S^N$ corresponding to different activities. The $a_S^N$ may be

located both within one $A_N$ or across as many $A_N$ as there are $a_S^N$. A minimum of one 'anchor' subarea is required per $O_C$ to identify a place of residence, $a_D^N$. For other activities (e.g. work or formal education) to be captured further $a_S^N$ are needed. Data driven assignment of occupants to subareas enables the exchange of $O_C$ by $A_N$ (Section 3.1). The 'anchor' $a_S^N$ are relatively static (i.e. changing infrequently) as for example, workplace remains constant for long periods.

If data do not allow direct matching of multiple 'anchor' $a_S^N$ for $O_C$, then  $a_D^N$ is assigned randomly (SciPy, 2019) but in proportion to the available choices. The choice can be informed by rules, such as imposed by local governing structures (e.g. school choice). For $O_C$ trips to non-anchor subareas (e.g. leisure activity, shopping), assignment is stochastic. Gravity weightings ($\Gamma$) for all potential trips between origin $i$ and destination $j$ locations ($B$, for coarser resolution than $A_N$) of distance $d_{i,j}$ are pre-calculated and stored in a matrix (Casey, 1955):

$$\Gamma_{i,j} = \frac{B_i \, B_j}{d_{ij}^2} \qquad\qquad (3)$$

where weights $\Gamma_{i,j}$ are derived by an attractor (e.g. total number of shops) within $B$ and the distance ($d$) between locations. The destination is randomly selected using gravity weightings (eq. 3), accepting amenity attraction rules (Reilly 1931). The process is nested to allow for spatial nesting of agents and account for spatial resolution of data on amenities.

Within an $A_N$ further rules, associated with movement, can be assigned to $O_C$ to represent structural and personal factors that impact timing and ability to move between $a_S^N$. For example, associated dependants (e.g. children) impact on timing of movement of an $O_C$ due to caring responsibilities.

### 2.3     Evolutionary dynamics

At each time step, the decision for an $A_N$ to release $O_C$ applies a Markovian approach (Appendix A). This stochastic state determination process decides the nature of an object's (e.g $O_C$) next state (e.g. $a_S^N$) using knowledge of its previous states (Blitzstein and Hwang, 2019). The subsequent time at which an $O_C$ is accepted by the destination $A_N$ is influenced by factors such as distance and time of travel. This allows random variability in human behaviour to be simulated such as presence and activities of occupants in a single building (Page et al., 2008; Richardson et al., 2008; Widén et al., 2009a) for long periods

(Page et al., 2008) whilst aggregate behaviour (informed social structure) will still be apparent. This requires knowledge (data) based on movement and location associated with time and allows decision making to be identified with individual $O_C$ as well as populations.

The movement and location data are used to create the Markov matrices' stationary distributions (eq. A1) for the exchange

of occupants at each time step ($t$). The Markov matrices are created prior to a model run but could be recalculated between each timestep of the model run in order to capture potential response (in movement and activity) to disruptions.

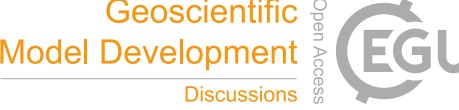

### 2.4    Calculation of $Q_F$

Heat sources (eq. 2) from people, buildings (with appliance load breakdown), and transport are determined using the $O_C$
count and associated activity in each of the $a_S^N$ of all $A_N$.

### 2.4.1    Metabolism $Q_{F,M}$

Metabolism ($Q_{F,M}$) of each $O_C$ uses an individual metabolic rate ($M$) as:

$$Q_{F,M,i} = M \cdot O_C \qquad (4)$$

with the sensible ($H$) and latent ($E$) components, using the Bowen ratio $\beta$ (sensible to latent heat) as (for one $O_C$):

$$Q_{F,M(E),i} = \frac{Q_{F,M,i}}{1+\beta} \qquad (5)$$

$$Q_{F,M(H),i} = \frac{Q_{F,M,i} \cdot \beta}{1+\beta} \qquad (6)$$

Both $\beta$ and $M$ can vary with activity (e.g. office work/sitting, walking) and demographics (e.g. age, gender).

### 220    2.4.2    Transport $Q_{F,T}$

If an $A_N$ releases an $O_C$, the journey time, route and mode of transport are needed to determine $Q_{F,T}$. These allow travel
dynamics to influence the time and nature of energy use at the associated spatial unit through a simple traffic model. $Q_{F,T}$ is
calculated at each timestep for the spatial units for each mode type $m$ (e.g. car, truck, train, walk) and route type $r$ (e.g.
minor- or major-road, over-ground- or below-ground-rail), with speed $v$ (m s$^{-1}$) and heat emission $F$ (W m$^{-1}$) for all travelling
$O_C$. The journey time is tracked to enable release of $O_C$ at appropriate (e.g. timely, delayed) periods at their destination $A_N$ by
using a mode and journey specific time bin ($t_b$). The journey time $t_b$ is updated at each time step. The notional duration is
found from the mode's distance/time relation using LOWESS analysis (Cleveland, 1988) on travel data for distance
travelled.

The total number of travelling $O_C$ in each spatial unit is the sum of $O_C$ in all $t_b$ for all $m$. The number of $O_C$ in a $t_b$ changes at
each timestep as, and when, new journeys begin. When the $t_b$ time is zero, the held $O_C$ are released to the next spatial unit of
their journey which may be a destination or an intermediate location (e.g. mode transfer from walking to bus).

The choice of $m$ is informed by data that associates probability of $m$ to origin-destination pairings. If journey combinations
data are unavailable, weighting by distance $d_{i,j}$ is used, informed by other sources (e.g. travel surveys). The journey route
(through different spatial units that calculate local $Q_{F,T}$) is determined from geographical information system (GIS) data (e.g.
OpenStreetMap contributors 2017), mapping application programming interfaces (APIs, e.g. Google (2019)) or straight line
distances between centroids (in the absence of data). For the latter, spatial nesting can be used between $A_N$ and $B$. Routing
options between spatial units can be one (most basic) or many (data dependent).


Route ($r$) parameters have a capacity limit ($R_{lim}$) assigned by $r$ related spatial (B, $A_N$) capacity constraints (e.g. size and
possible number of occupants of a bus or a railway carriage that operate in that area, road congestion limits). However, these
may be modified if a disruption impacts part of the transport network (e.g. power failure, intense flooding). The current
occupancy is constrained by a mode-appropriate ratio ($C_{m,r}$) such as number of occupants ($n_{o,m,r}$) per unit vehicle. For road
related transport, unit vehicle length ($L_m$) is required as, for example, buses hold more people than a car but require more
space on the road. These constraints are informed by local data.





A total vehicle count for each $m, r$ (as $V_{m,r}$) is used to determine if $O_C$ in travel can be moved between spatial units. When both

$\qquad V_{m,r} \leq \frac{n_{o,m,r}}{c_{m,r}}$ and $\left( \left( \sum_{m=1}^{lim} V_{m,r} \cdot L_m \right) + \Delta_{V_{m,r}} \right) \leq R_{lim}$ (7)

then $V_{m,r}$ is incremented by $\Delta_{V_{m,r}}$ (i.e. $V_{m,r} + \Delta_{V_{m,r}}$) where $\Delta_{V_{m,r}} = \frac{O_C}{c_{m,r}}$. If $R_{lim}$ (e.g. total road-type length in a spatial unit) is exceeded, $O_C$ will not be passed to the next spatial unit: time associated ($t_b$) in neighbouring spatial units will be lengthened. When

$\qquad V_{m,r} > \frac{n_{o,m,r}}{c_{m,r}}$ (8)

then $V_{m,r}$ becomes $V_{m,r} - \Delta_{V_{m,r}}$.

Where transport is considered at the spatial resolution of $B$, $V_{m,r}$ are distributed to child spatial units based on the ratio of nested spatial unit capacity to the parent spatial unit's capacity (e.g. $L_{m,A_N}/L_{m,B}$ for cars).

The anthropogenic heat flux from transport, $Q_{F,T}$ for an $A_N$ of area A, at time $t$ is (Grimmond, 1992):

$\qquad Q_{F,T} = \frac{\sum_{r=1}^{nr} \sum_{m=1}^{nm} V_{m,r} \cdot F_{m,f} \cdot L_{r,t}}{A}$ (W m$^{-2}$) (9)

where $L_{r,t}$ is the distance travelled in a time-step. Heat emission ($F_{m,f}$; W m$^{-1}$) varies with fuel type ($f$), $m, r$ and vehicle speed ($v_{m,r}$; m s$^{-1}$). For the case of road traffic, speed can be represented as a function of permitted, or average speed limit ($v_{r,lim}$). This is linked to traffic density (i.e. vehicles per unit length, e.g. Salter, 1989) which we relate to a ratio of total on-road

vehicle length to total route length (equates to $R_{lim}$) as:

$\qquad D = \frac{\sum_{m=1}^{lim} V_{m,r} \cdot L_m}{R_{lim}}$ (10)

Hence, speed-density function changes with time (e.g. Greenshields et al. 1935; Wu 2000):

$\qquad v_r(t) = v_{r,lim} - D(t) \cdot v_{r,lim}$ (m s$^{-1}$) (11)

The relation of $v_r(t)$ to $F_{m,f}$ is dependent on local fuels types (e.g. Grimmond, 1992; Smith et al., 2009) and is part of the

model parameters specification (e.g. Section 3).

### 2.4.3 Building energy ($Q_{F,B}$)

$Q_{F,B}$ accounts for appliance usage ($Q_{F,B}^{\alpha}$), lighting ($Q_{F,B}^{l}$), heating and cooling demands ($Q_{F,B}^{HC}$) and hot water demand ($Q_{F,B}^{HW}$):

$\qquad Q_{F,B} = Q_{F,B}^{\alpha} + Q_{F,B}^{l} + Q_{F,B}^{HC} + Q_{F,B}^{HW}$ (W m$^{-2}$) (12)

These vary by $A_N$ as $O_C$ composition changes activities $a_S^N$, and the local building form, construction (materials and dimensions), and control systems (heating, cooling, lighting) change (e.g. as neighbourhood age or construction period varies). $A_N$ release (acceptance) of $O_C$ to (from) the movement and travel module leads to a change in occupancy levels in associated building types. Activity of $O_C$ informs appliance ($\alpha$), hot water (HW) and lighting ($l$) energy use as well as heating and cooling (HC) set-points for building environmental control.


$Q_{F,B}$ is determined through use of a **s**implified **t**hermal **e**nergy **b**alance for **b**uildings **s**cheme (STEBBS) that calculates heat transfer through building fabric and ventilation using an adjustable time resolution. $Q_{F,M}$, $\alpha$, HW, and $l$ provide internal gains to the building volume and fabric (Appendix B). The dynamic 1-D energy model enables both simple representation of individual buildings (Klein et al., 2017), as well as scaling to represent groups of building within an $A_N$. By using building

archetypes, STEBBS provides a computationally efficient representation of buildings across a city (Heiple and Sailor, 2008; Bueno et al., 2012; Kikegawa et al., 2014) and permit multiple types within an $A_N$.



For each archetype with an $A_N$, STEBBS requires the building dimensions (width, depth, height), window-wall ratio, and thermo-physical properties for the building components (i.e. window, wall, roof, floor, internal mass). Thermal inertia of appliances and lighting is assumed to be negligible (i.e. no regulating thermal mass) and so the heat resulting from their use (i.e. total power demand $P_a$) is exchanged directly with the indoor air.

Domestic hot water (DHW, following building services convention this includes both domestic and commercial buildings) heating and air heating/cooling are a response to internal conditions, controlled by a setpoint temperature ($T_{set}$; K). The energy use ($q$) depends on the system efficiency ($\kappa$) and maximum power rating ($P_{max}$) for heating using an exponential control to avoid heating overshoot:

$$q_H = \kappa \left( P_{max} - \frac{P_{max}}{exp^{(T_{set} - T_i)}} \right) \qquad \text{(W)} \quad (13)$$

and for cooling:

$$q_C = \kappa \left( P_{max} - \frac{P_{max}}{exp^{(T_i - T_{set})}} \right) \qquad \text{(W)} \quad (14)$$

where $T_i$ is the internal water/air temperature (K). Efficiency losses of the heating system and all cooling energy are calculated as direct heat ejection to the outdoor environment. The heating of the building fabric modifies the storage heat flux of the urban energy balance (Grimmond et al., 1991; Grimmond and Oke, 1999). Thus this term is tracked and removed from $Q_{F,B}$.

Ventilation loss/gain ($q_{vent}$) is given as (Spitler, 2011):

$$q_{vent} = V_R \, \rho_a \, c_p (T_o - T_i) \qquad \text{(W)} \quad (15)$$

where $V_R$ is the ventilation rate (m$^3$ s$^{-1}$), $\rho_a$ is the air density (kg m$^{-3}$), $c_p$ is the specific heat capacity of air at constant pressure (J kg$^{-1}$ K$^{-1}$), and $T_o$ the outdoor air temperature (K). In the standalone version of this model no spatial variations of these are considered. If coupled to a meteorological model these outdoor variables can be spatially dynamic and respond to $Q_F$ emissions locally (Sun and Grimmond, 2019).

DHW is considered as a sensible heat gain only (no latent) with hot water to drains unaccounted for in $Q_{F,B}$. Heat exchange between DHW in storage (tank and water pipes) and building volume is accounted for. Volumetric flow rates ($V_{FR}$, m$^3$ s$^{-1}$) of DHW use and to-drain can be set to control volume of DHW in-use. The internal heat gain from this varies with $O_C$ level and activity.

The combined internal gains based on internal building activities are passed to STEBBS. The number of active (i.e. present and awake) $O_C$ in a building (e.g. domestic, work) influences total energy use (Druckman and Jackson, 2008; Yohanis et al., 2008) and the energy demand profiles at timescales from seconds (Richardson et al., 2010) to hours (Widén et al., 2009b). Hence, occupancy levels are essential to reproducing commercial (Kim and Srebric, 2017) and domestic load patterns (Widén and Wäckelgård, 2010).

Hence, each building archetype within an $A_N$ is impacted by its $O_C$ level and their activities (i.e. $a_S^N$). As $O_C$ categories (e.g. age related) participate in different activities (e.g. infant differs from adult), local census (or other) data both constrain and spatially inform $O_C$ characteristics.

Lighting and appliance gains are associated with activity, appliance type $\alpha$ (Firth et al., 2008) set efficiency and power usage ($P_\alpha$) associated with different building types (e.g. commercial, domestic). We distinguish three energy consumption classes:

(i) *active only (AO)* - only occurs with user activity (e.g. oven, iron)





(ii)   *continuous (C)* - always consuming energy (e.g. cold appliances: fridge, freezer; small appliances: telephone, clock, burglar alarm). As these may cycle power (e.g. cold appliances) the power rating accounts for the fraction of time the appliance draws power during a single complete cycle and the mean power consumed whilst operating.

(iii)   *active/standby (AS)* – two modes which depend on user activities (e.g. television, computer): (1) as *AO,* (2) less when not actively used.

Each appliance ($\alpha$) type ($j$) is assigned to either *AO, C,* or *AS* with an active power rating $\alpha_p$ and additionally for *AS* appliances a standby rating $\alpha_s$. The number of appliances of type $j$ in $A_N$ ($\alpha_{j,A_N}$) is determined by domestic/non-domestic appliance market permeation ($\alpha_{j,k}$) as:

$$\alpha_{j,A_N} = \alpha_{j,k} \cdot n_b \qquad (16)$$

where $n_b$ is number of households (domestic), number of work-desks (non-domestic, commercial), or floor area (non-

domestic, other) in an $A_N$. $\alpha_{j,A_N}$ acts as the limit of appliance use at any time. If no distinction between $j$ use profiles can be given (data dependent) all appliance demand is combined as one type.

For domestic use, households are categorised by total number of residents such that proportion of $\alpha_{j,A_N}$ (by *AO, C,* or *AS*) in use at a given time $t$ is:

$$\alpha_{j,u}(t) = \sum_{x=1}^{lim} f_{x,\alpha_j}(t) \cdot \frac{n_{b,x}(t)}{n_b} \cdot \alpha_{j,A_N} \qquad (17)$$

with $f_{x,\alpha_j}(t)$ the fraction of households with $x$ active occupants using $\alpha_j$ at $t$ (based on occupant activity scheduling) and $n_{b,x}(t)$ the number of households with $x$ active occupants at $t$. For non-domestic buildings, appliance use is proportional to occupancy level and lighting is considered part of this load.

The power demand $P_\alpha$ (W) of all appliances in use is:

$$P_\alpha = \underbrace{\sum_{j=1}^{n_{AO}}(\alpha_{j,u}(t) \cdot P_{\alpha_j}) + \sum_{j=1}^{n_{AS(1)}}(\alpha_{j,u}(t) \cdot P_{\alpha_j})}_{P_{\alpha(active)}} + \underbrace{\sum_{j=1}^{n_{AS(2)}}[(\alpha_{j,A_N} - \alpha_{j,u}(t)) \cdot P_{\alpha_j}]}_{P_{\alpha(standby)}} + \underbrace{\sum_{j=1}^{n_C}(P_{\alpha_j} \cdot \alpha_{j,A_N})}_{P_{\alpha(continuous)}} \quad (18)$$

and is the heat gain passed to each STEBBS instance (i.e. each building archetype per $A_N$). Appliance characteristics are currently uniform throughout $A_N$ but could be variable (e.g. by socio-economic structure).

Domestic lighting is considered as a separate load impacted by an outdoor downwelling shortwave radiation threshold ($K_{\downarrow lim}$), number of households with active (awake) occupants $n_{b,x}$; and a base/min/max luminous intensity, $l_{base/min/max}$, per household for scaling lighting requirement (Widén et al., 2009a):

$$K_\downarrow(t) < K_{\downarrow lim}: P_{light} = P_l \cdot n_{b,x} \cdot \left[ l_{base} + \left( l_{min} \cdot \frac{K_\downarrow(t)}{K_{\downarrow lim}} + l_{max} \cdot \left(1 - \frac{K_\downarrow(t)}{K_{\downarrow lim}}\right)\right)\right] \quad (W) \quad (19)$$

Luminous intensity is converted to total power ($P_{light}$) using a per light power rating ($P_l$). This is passed to STEBBS as part

of the appliance load $P_\alpha$.

## 3   Evaluation of DASH in Greater London

### 3.1   DASH setup and data sources

We evaluate DASH in Greater London (GL). In the United Kingdom (UK), the smallest spatial unit that census data are

provided for is the Output Area (OA). We adopt the OA as the agent spatial units (i.e. $A_N$) in the model runs with $A_N$ nested within four coarser spatial units ($B$): Lower-level Super Output Area (LSOA); Mid-level Super Output Area (MSOA); Local Authority (LA); and City/Region as data (from various agencies) are aligned to one or more of these spatial units. The LA have several governance roles (e.g. traffic speed, school districts, planning decisions, etc.) that will impact energy use (LGA, 2019). Similar structures are used in other countries but with varying levels creating the complete city (National Bureau of





Statistics of China, 2017; Statistics Bureau of Japan, 2017; Statistics Canada, 2017; US Census Bureau, 2019). In London there are 25,053 OA (determined by residential population and social homogeneity, Office for National Statistics (2017a)) that vary in size from $1.56 \cdot 10^{-4}$ to 12.3 km$^2$, 4,835 LSOA, 983 MSOA, and 33 LA within one Greater London Authority Region (Table 1).

The UK Time Use Survey (TUS) 2014 – 2015 (Gershuny and Sullivan, 2017) provides a structured source of data for simulating population movement and human activity (Iamarino et al., 2012; McKenna et al., 2015; Baetens and Saelens, 2016). Such surveys are carried out in many countries by governments or research institutes (Fisher and Gershuny, 2013), allowing DASH to be applied elsewhere with appropriate cultural practises accounted for. In the UK TUS, residents record their activities and location for one weekday and one weekend day, normally creating profiles of individuals with income,
age, sex and household type meta-data. The data samples are sufficient to allow analysis at national to regional (e.g. GL) scale in many cases. The 10 min time-step resolution of TUS data (Gershuny and Sullivan, 2017) is the basis for the model timestep.

      The TUS data are used to construct Markov chains (Appendix A) that govern the exchange of occupants in DASH (Fig. 1a)
and the levels and type of activities undertaken by different groups of $O_c$ across the day (section 2.3, Table 2). Age cohorts (Table 2) are used as the group identifier. Appliances attributed to TUS activities (Table 2) have different power ratings and market permeation (Table 3, C1). Non-domestic activity varies by workplace appliance types according to the land use (e.g. industrial, office) of the $A_N$ (BEIS 2017a; OpenStreetMap 2017).

The application is undertaken for 2015 to coincide with the TUS data, when GL had a population of 8.539 million (census data updated annually, Table 2). The remaining data needed are obtained for the closest year. Throughout we endeavour to use open-source, freely available data. A variety of data types are used, at a range of spatial resolutions (Table 1) with more detail given subsequently (Table 2-5).

Movement of occupants is informed by the National Travel Survey (DfT, 2017) and census data on commute patterns ($\S_{8,10}$, Table 4), to determine choice of mode by distance or type of journey providing the travel attributes (Table 4). In this evaluation, nine modes of transport ($m$) exist: cars, motorcycles, vans, taxis, buses, surface rail, underground rail, cycling and walking. Other deployments could include freight and boat related modes. Exclusion of freight vehicles does not directly affect the travel dynamics, but will result in an underestimation of $Q_{F,T}$. Route types ($r$) considered, include four road types
(residential, minor (so called B-roads in the UK), major (UK's A-roads) and motorways (highways)); and two rail types (underground and surface). In the model runs, journey distances for all routes that move between LAs are determined at LA scale based on GIS shapefile LA centroids. This is the coarsest implementation of the transport component of the model.

      STEBBS is used with different parameters for domestic and non-domestic buildings (Field, 2008). We simplify to the three
most common domestic building (houses, bungalows, and flats) archetypes in GL, varied by presence at LSOA level (Table 3, Mavrogianni et al. 2012; Valuation Office Agency 2015). Despite advances in non-domestic buildings characterisation for GL (Evans et al., 2019), the heterogeneity in form and use limits use of a range of archetypes (Steadman et al., 2000). Again, for simplicity in this evaluation, we use a single STEBBS characterization based on the most common domestic archetype parameters for non-domestic (e.g. shops, hospitals, offices). Hence, a maximum of four STEBBS instances per $A_N$ with the
appropriate building fabric thermo-physical properties assigned from one of two building age groups (pre- or post-1965, Tables 3 and C2). Building dimensions are informed by total $A_N$ building footprint and height (Table 3) for each archetype





by age category. The limited consideration of building material thermophysical properties is expected to reduce the spatial variance in heating and cooling contributions to $Q_F$ in DASH.

Meteorological data to force the model are from the KSSW site in central London (Kotthaus and Grimmond, 2014, Table 5). Means (1 and 5-min) are used to obtain 10-min means (model time step). Outgoing longwave radiation observed with a Kipp and Zonen CNR4 radiometer (Table 5) is used assuming an emissivity of 0.9 (Butcher and Craig, 2016) and Stefan-Boltzmann equation (Oke, 1988) to obtain surface temperature. Soil temperature (at 5 m depth) is assigned assuming it is equivalent to the mean annual (2014-2015) air temperature (Sellers, 1972; Busby, 2015) of 11.9 °C.


As the model requires continuous atmospheric data, gaps are filled in consecutive order: (a) linear interpolation when less than 4 h; (b) median for same time in the surrounding $\pm$ 48 h for gaps of 4 – 24 h; and (c) similarly for gaps greater than 24 h, using the median $\pm$ 72 h. The various model runs (Table 6) have a spin-up period of 24 h (144 timesteps) for the STEBBS model to become stable.


### 3.2    Evaluation methodology

Ideally a model is evaluated with observations of the simulated variables (Table 6). However, direct observations of $Q_F$ are extremely limited or are indirect with a series of assumptions within them. At the neighbourhood scale, combining radiation and eddy covariance observations while assuming energy balance closure has been used to assess monthly and daily values

(e.g. Offerle et al., 2005; Pigeon et al., 2007). Using satellite earth observation, a much larger spatial extent (e.g. city wide) is observed but with a bias to clear sky conditions. The snapshot values at the time of the satellite overpass require a very large number of assumptions in addition to energy balance closure (e.g. Chrysoulakis et al., 2018). The closest to "direct" measurements of $Q_F$ are micro-scale emissions from building vents (i.e. part of $Q_{F,B}$) using eddy covariance sensors (Kotthaus and Grimmond, 2012) but there are extremely limited data available. Thus, the spatial and temporal scales that

DASH is capable of simulating cannot be directly compared to measured $Q_F$. We therefore use a series of different sources of public data and another model to evaluate various aspects of DASH.

The reference model used, GQF (Iamarino et al., 2012; Gabey et al., 2019), is a top-down inventory $Q_F$ model developed for London. This is selected as it is amongst the most (spatially and temporally) detailed models for London currently available

(Gabey et al., 2019). We apply it to 2014 – 2015 to align with metered data used in the evaluation. The model uses energy consumption, traffic, and workday population data to provide half-hourly estimates of $Q_F$ at city, LA, and OA resolutions. Hence, $Q_F$ estimates for both models are at city scale with OA resolution.

There are several GQF features that restrict DASH being evaluated at higher detail. These are: (i) GQF uses data from a

range of scales (up to national) to determine OA results with population weighted disaggregation; (ii) diurnal patterns are prescribed based on either assumptions or coarse spatial data, with variation by day type (weekday, weekend) and season – meaning variability at smaller scales are not captured; (iii) GQF assumes the same diurnal profile for both gas and electricity usage; and (iv) effects of temperature in GQF are the net seasonal diurnal energy use profiles rather than reproducing the day-to-day conditions in London. Hence, individual DASH diurnal patterns cannot be evaluated against GQF with fine

temporal or spatial resolution as differences are expected.

To evaluate DASH, appliance (including cooking) power demand is equated to GQF electricity demand and DASH heating and cooling demand to GQF gas demand. This will lead to discrepancies as the demand profiles used in GQF are not energy



carrier or vector specific. The calculation and evaluation of $Q_{F,T}$ is undertaken at $A_N$ scale rather than individual routes. In both models, many of the minor residential roads in $A_N$ are unaccounted for.

DASH evaluations (Table 6) use annual (1 Oct 2014 to 30 Sept 2015) publicly available gas and electricity consumption data (GWh) for domestic and non-domestic (commercial + industrial) use (BEIS 2017a,b) and national gas transmission operational data for the same period(NG, 2015). DASH, run with the appropriate meteorology (Table 5), OA results are aggregated for assessment to the LSOA (domestic) and MSOA (non-domestic) scales. These evaluation data have some issues: (i) some non-domestic meter data are undisclosed at MSOA level but appear at LA level (without a MSOA) (BEIS 2018); (ii) meters with insufficient address metadata cause underreported consumption statistics for some areas; (iii) some gas consumption statistics may be wrongly classified (domestic/non-domestic) as this is done based on annual consumption (threshold =73200 kW h year$^{-1}$) (BEIS 2018); and (iv) spatial misallocation of metered commercial gas consumption to the billing address rather than actual building/location of use (BEIS 2018).

Basic metrics assessed include the median (50%), interquartile range (IQR), and standard deviation (SD). To evaluate the modelled ($X_{M,i}$) and observed (or reference) ($X_{O,i}$) time and/or spatial data series both the difference:

$$\Delta_i = X_{M,i} - X_{O,i} \qquad (20a)$$

and the absolute errors

$$AE_i = |\Delta_i| \qquad (20b)$$

are determined, from these:

(1) Cumulative distribution of $AE_i$ is obtained all values (e.g. across all 25,053 OA, Fig. 9)

(2) Normalised by maximum: $nMax = \dfrac{X_i}{\max(X_i)}$ (e.g. Fig. 10)

(3) Normalised errors (%): $nE_i = (\Delta_i/X_{O,i})100$ (e.g. Fig. 11a,b, ideal value would be 0).

(4) Absolute normalised error: $AnE_i = \left| \dfrac{X_{M,i}}{\max(X_{M,i}) - \min(X_{M,i})} - \dfrac{X_{O,i}}{\max(X_{O,i}) - \min(X_{O,i})} \right|$ (e.g. Fig. 11c, d, ideal value would be 0).

## 4    Analysis of model dynamics

As behaviour, demographics, and travel choices influence the temporal and spatial variation in movement and activity profiles in DASH $Q_F$ estimates, we examine these first. A critical control on $Q_F$ is the number of occupants within an area. The area itself may be static (e.g. where buildings are located) or moving (e.g. transport area). The occupancy level will change as people travel to different locations (Fig. 2).

In model run R1 (Table 6), the results for one B spatial unit (LA Camden, London) are used to demonstrate the $O_C$ movement and travel through time (six consecutive days) within each $a_S^N$ for each age group for three day types (weekday (school/non-school), weekend) as a result of $A_N$ occupant exchange (Section 2.2). The occupancy levels vary by day type and between age groups, whilst having general consistency within day-type by age cohort. Note, people travel outside (and into) this B during the period, but no perturbation is undertaken (e.g. changing transport availability or road construction).

During school weekdays most children and teenagers are in school ($a_E^N$, $a_H^N$). Adults, some teenagers, and some seniors work during all day types, and during all times of day. Adult $a_W^N$ occupancy at work (increase at home) is slightly lower on non-school (NS) weekdays than school/work (SW) days as a result of childcare - a small dip observed during noon on NS and SW days that reflects lunchtime activity. $a_D^N$, $a_R^N$ and $a_O^N$ occupancy levels increase after peak school and work times, with $a_D^N$ occupancy returning to similar levels each night.





The occupancy levels of each $a_D^N$, $a_W^N$, $a_E^N$, $a_H^N$ are partly informed by population data, so it is important realistic values occur from the movement processes. This is assessed by comparison of the median and IQR of the total occupancy across each $a_S^N$

in the city to the static populations of each $A_N$ and subarea (i.e. residential, workday, school populations) for one weekday (Fig. 3). Hence, a value of 1 indicates the total population is present. $a_W^N$ occupancy levels have a median peak just over 0.6 of the workday population. $A_N$ interaction in DASH allows for different types of work, such as full/part-time and shift work, as it is inherent to the movement data (in this case the TUS, Table 2). Whilst this might not reflect the accurate behaviour of a particular $a_W^N$ (e.g. an $a_W^N$ comprising entirely office work may in reality only be occupied 09:00-17:00), the total

variability over a group of $a_W^N$ may be more realistic, given varying work times between commercial sectors.

For R2 (Table 6) both $a_E^N$ and $a_H^N$ IQR occupancy levels are less than some $A_N$ school populations (Fig. 3), but for morning to noon $a_H^N$ the population is exceeded in some areas. Both the deficit and surplus may relate to the method of assigning school anchors to child and teenager $O_C$ (Section 2.2). If the age group residential population is lower (higher) than the school

population in a LA, there will be too few (many) students occupying this LA schools during the day. As students are assumed not to cross LA boundaries, given state school catchment area restrictions. In Greater London 89% of pupils are in state schools (DfE, 2019).

$a_D^N$ occupancy levels are always below 1. The highest values occur overnight when most people are expected to be at home.

The narrow IQR indicates there is little variation in total occupancy levels between areas. Variations are expected with active occupancy (e.g. household sizes, Section 2.3.1) and in $a_D^N$ with large differences in resident age groups.

Total occupancy varies with behaviour of different age groups and will affect the power demand within the neighbourhood. To demonstrate the impact of demographics on daily profiles of $O_C$ in the $a_D^N$, three $A_N$ (neighbourhood, OA, scale) with

similar residential populations but different dominant age-cohort are compared in Fig. 4 (R3, Table 6). The $a_D^N$ of each of the three $A_N$ have distinct dominant age groups as: $a_D^{senior}$ 78% (291) residents are seniors; $a_D^{working}$ 92% (297) residents are adults; and $a_D^{young}$ 47% (300) residents are infants, children or teenagers. In $a_D^{senior}$ (Fig. 4a), daytime $O_C$ remains proportionally higher (Fig. 4d) than $a_D^{working}$ (Fig. 4b) and $a_D^{young}$ (Fig. 4c). $a_D^{young}$ has a steeper morning decrease in $O_C$ and earlier inflection point in the afternoon than $a_D^{working}$ , likely due to formal school day lengths (Fig. 2). On the weekend

day, all age groups, apart from teenagers, follow similar patterns, with about 60 – 70% remaining in the $A_N$ (Fig. 4d).

The diurnal pattern of occupancy levels by day type is consistent between days and boroughs (R4, Table 6). The variability of borough occupancy levels for $a_D^N$ (Fig. 5a) and $a_W^N$ (Fig. 5b) is greater in the daytime when movement is more likely. Although, these standard deviations are quite small compared to the actual LA-level residential (8,760 - 379,691 residents)

and workday (58,444 – 356,706 workers) populations (ONS, 2014a, 2015). This demonstrates that the occupancy exchange method (Section 2.2) produces variation in occupancy levels on a daily basis when the same parameters are used for each day.

In this road vehicle evaluation (R5, Table 6), routing is at LA scale with inter-LA routes determined using Google

Directions (Google, 2019). The volumes of vehicles in use by mode (Figure 6) predicted by the movement component (Fig. 1, Section 2.3) peaks in the morning (07:30-09:30). Slight increases are present around noon and early evening. Low values (00:00-06:00) occur when movement is low (Fig. 2). The increase at 04:00 is due to both low sampling and the temporal





boundary of the TUS, which considers a day's worth of entries to occur 04:00-04:00. The volume of buses is constant over

the period 08:00 – 20:00 due to an imposed condition on capacity that represents an increase in $C_{bus,r}$ (Section 2.4.2) instead

of increasing $V_{bus,r}$. With only one route option given per LA origin-destination pair, road traffic is distributed between $A_N$ in

proportion to LA total road area. Routing options at $A_N$ scale have not been implemented.

## 5     Evaluation of DASH with GQF

The DASH evaluation assumes average or typical conditions (i.e. no disruptions are imposed to modify movement and/or

timing of activity). As a result the contribution of appliance use to $Q_{F,B}$ is expected to be similar for all days of each type

(e.g. weekday, weekend) throughout the year for both domestic and commercial settings (seasonality in appliance-based

activity is not considered). In a non-perturbed state, variation within day types across a year is expected to come from

heating (space and water) and cooling use as these demands respond to immediate environmental forcing within DASH. As

GQF (section 3.2) only varies electricity demand with day type and season and gas with season, we compare the DASH

diurnal pattern and magnitude of $Q_{F,B}$ components for two school weekdays (SW) in different seasons (summer: 18 June

2015, winter: 27 January 2015). The mean air temperature is warmer in summer (17.0°C) than winter (7.0°C) and has more

total radiation (Fig. 7).

To evaluate heat emissions from buildings ($Q_{F,B}$) the city-wide emissions of domestic (dom) and commercial/non-domestic

buildings (n-dom) are considered separately (R6, Table 6). As DASH and GQF have the same spatial resolution, comparison

is made of spatial inter-quartile ranges (IQR) at the GQF 30-min temporal resolution (i.e. 30-min means (time-ending) are

calculated from the DASH 10-min values). DASH appliance emissions ($Q_{F,B}^{\alpha}$) are compared to GQF electricity demand

($Q_{F,B}^{elec}$) whilst combined heating (space and water) and cooling ($Q_{F,B}^{HC} + Q_{F,B}^{HW}$) in DASH are equated to GQF gas demand

($Q_{F,B}^{gas}$). Discrepancies between values are expected, for example in some areas heating may be powered by electricity.

For the summer weekday, DASH domestic $Q_{F,B}$ has similar characteristics to GQF with consistent morning and evening

peaks. The mean and IQR are similar from midnight to 5 am, but consistently lower (difference in medians of 2 – 2.5 W m$^{-2}$)

in DASH from the morning to end of evening peak (Fig. 8ai). Across spatial $A_N$ more than 60% have an absolute error (AE,

eq. 20b) of ≤ 2 W m$^{-2}$ for all times sampled, and for ~90% the AE ≤ 5 W m$^{-2}$ (Fig. 9a).

Domestic $Q_{F,B}^{\alpha}$ closely follows $Q_{F,B}^{elec}$ in both pattern and magnitude on the summer day. DASH has three distinct appliance

demand peaks: morning, midday, and a larger more sustained evening peak. The magnitude and timing of $Q_{F,B}^{\alpha}$ and $Q_{F,B}^{elec}$

peaks are similar between DASH and GQF, although the morning peak in GQF is maintained with less variability

throughout the day (Fig. 8a.ii). The domestic summer day gas (GQF) and heating/cooling (DASH) $Q_{F,B}$ profile (Fig. 8a.iii)

have the largest discrepancy in daily profile and magnitude. Under summer conditions, DASH heating/cooling is largely

driven by hot water demand as indoor temperatures in all instances of STEBBS are passively maintained between heating

and cooling setpoints.

DASH domestic $Q_{F,B}$ has a more distinct morning peak in winter (Fig. 8di), and from midnight to the morning peak DASH

values are 1 – 4 W m$^{-2}$ greater than GQF. This is caused by greater $Q_{F,B}^{HC+HW}$, and may relate to greater sensitivity to

temperature for DASH and low outdoor air temperatures. The evening peak is less pronounced and shifted to later evening,

with roughly 70% of the $A_N$ having AE ≤ 5 W m$^{-2}$ at 18:00 (Fig. 9b). All other times analysed more in agreement with GQF.

$Q_{F,B}^{\alpha}$ remains similar to the summer values (Fig. 8.aii) as the only seasonal variation is due to indoor lighting. After the





morning peak it is slightly lower than $Q_{F,B}^{elec}$ (Fig. 8d.ii), but follows a similar pattern throughout the day. This discrepancy is likely due to electric heating use, which $Q_{F,B}^{elec}$ would include on both a small (e.g. space heaters) and large (e.g. 'district' electric heating in high-rise flats) scale.

Summer commercial $Q_{F,B}$ is consistently lower in DASH (median ~1.5 W m$^{-2}$ less) than GQF in the middle of the day (Fig.

8b.i) with morning and evening medians more similar. The evening IQR increases for DASH and is reflected in $Q_{F,B}^{\alpha}$, likely associated with energy demand from commercial properties that remain open later in the evening (e.g. leisure facilities). There is close agreement between $Q_{F,B}^{\alpha}$ and $Q_{F,B}^{elec}$ medians (Fig. 8b.ii). At least 60% of $A_N$ agree within 2 W m$^{-2}$ for all sampled time steps (Fig. 9c).

The winter diurnal patterns for commercial $Q_{F,B}$ are similar for DASH and GQF (Fig. 8e.i) but DASH has a steeper morning (evening) increase (decrease) as well as consistently higher values (median 2 - 3 W m$^{-2}$ in the daytime). The evening decrease starts ~ 2 h later in DASH. These higher values are due to $Q_{F,B}^{HC+HW}$ (Fig. 8e.iii), which dominates the total pattern. The median $Q_{F,B}^{\alpha}$ and $Q_{F,B}^{elec}$ profiles (Fig. 8e.ii) are in good agreement, with slightly broader IQR for DASH. More than 50% of $A_N$ have a MAE of ≤ 2 W m$^{-2}$ for all times except 09:00, which is slightly below 50% (Fig. 9d).


For both domestic and commercial use, summer $Q_{F,B}^{HC+HW}$ have the largest discrepancy in profile and magnitude compared to $Q_{F,B}^{gas}$ (Figs. 8a.iii, 8b.iii). In summer for DASH, $Q_{F,B}^{HW}$ is expected to dominate as indoor temperatures in all instances of STEBBS are passively maintained between heating and cooling setpoints. City-wide domestic $Q_{F,B}$ is greater than commercial $Q_{F,B}$ in both DASH and GQF.


The median $Q_{F,T}$ are fairly similar between both models but GQF has less temporal variability (Figs. 8c.i, 8f) with IQR$_{DASH}$ ~ 4 x IQR$_{GQF}$. As DASH responds to variations in travel demand, and exchanges occupants across the city more temporal variation occur between $A_N$. Figs. 9e, f, show small MAEs between the two models, with more than 98.5% of $A_N$ within 2 W m$^{-2}$. When considered for road area only, DASH $Q_{F,T}$ median values reach 2.9 W m$^{-2}$, with diurnal mean of 3.25 W m$^{-2}$ (Fig.

8c.ii). Summer (Fig. 8c.i) and winter (Fig. 8f) values differ because of the behavioural change caused by daylight savings time. But no other seasonal changes are expected or occur.

Here the mean GQF values are based on key day types appropriately weighted for the year, whereas DASH is run for the year. The GL annual average $Q_{F,M}$ for DASH is 0.663 W m$^{-2}$, for GQF it is 0.717 W m$^{-2}$, whereas assuming one mean

metabolic flux for all that live in GL gives 0.386 W m$^{-2}$. The GL annual average $Q_{F,T}$ from DASH (0.24 W m$^{-2}$) is larger than for GQF (0.0303 W m$^{-2}$) as GQF uses a smaller road network (OS (2016) vs. AADT, respectively). The GL annual average $Q_{F,B}$ for DASH (5.74 W m$^{-2}$) is slightly smaller than the 2015 average meter data (7.22 W m$^{-2}$, Section 6). The GL annual total $Q_F$ for DASH (6.0 W m$^{-2}$) is smaller than for GQF (7.97 W m$^{-2}$). The Iamarino et al. (2012) (earlier version of) GQF annual average (10.9 W m$^{-2}$) for 2005 to 2008 is larger which is consistent with the decrease in published values seen for

London (e.g. Ward et al., 2016; Ward and Grimmond, 2017).

## 6    Evaluation of DASH with annual gas and electricity consumption data

To assess the annual DASH city-wide hot water, heating and cooling energy demand (R7, Table 6) results are compared to normalised national gas demand. The seasonal pattern (winter peak, summer minimum) is evident in both (national, DASH) heating data, with short and long period responses to temperature also evident (Fig. 10). The DASH response to the higher

frequency variations is similar to the demand data but the amplitude of normalised demand differs. DASH is seemingly more



sensitive to temperature changes but as the national demand profile has net local responses to weather (*etc.*) variations across the country these may be smoother than if only London responses were observed.

In June to August, DASH heating/cooling demand is solely attributed to DHW demand for both domestic and commercial buildings. The consistency in DASH daily-behaviour (i.e. R7 without imposed perturbations) results in a steady-state summer load, with a baseline demand that is less dependent on environmental variability. The normalised national data have both greater magnitude and amplitude of fluctuation in summer (cf. DASH). The national data includes appliance (e.g. cooking) and industrial gas demands whereas DASH accounts for these in appliances (omitted in Fig. 10). The heating season dominates the DASH results (Fig. 10). The DASH pattern is less variable with the cooking and industrial baseline

demands included (not shown).

Evaluation of DASH (R7, Table 6) at LSOA scale (Table 1) suggests the DASH total domestic energy consumption is less than metered values (Fig. 11a.i). The DASH IQR is 46 to 29 % lower (Fig. 11a.ii). Although the LSOA domestic consumption in the central business district (CBD – City of London) has the largest

discrepancy (-81.5%), this may be due to misallocation in the published data (e.g. some dwellings classified as commercial because of a large shared meter). There is no evidence of a relation between percentage difference and population density.

The percentage difference between commercial DASH and non-domestic energy consumption is skewed to overestimation

by DASH in most MSOAs (Fig. 11b.ii). The CBD underestimation (-52.9%, Fig. 11b.i) is likely caused by a large misallocation of commercial gas consumption in this area (section 3.2). Two spatial units (West and East London) overestimate by more than 1000% (maximum being 1150%, 25.5 GW h). Some OAs (i.e. $A_N$ scale) with large retail buildings have potential uncertainty in both the energy consumption data (e.g. undisclosed data, section 3.2) and DASH simulations.

At MSOA scale, DASH simulates 38% of the areas to within ±100% of published values. The MSOAs that DASH most overestimates (as percentage differences) have fairly small actual magnitude differences and low workplace populations. The mean difference in magnitude across the top 5[th] percentile is 28.7 GW h, however 77% of these (mean difference 18.1 GW h) have workday populations of fewer than 2,000 people in the MSOA, with most businesses in these MSOA having fewer

than 50 employees. Whilst the proportion of these small businesses is fairly high (89% on average) across Greater London (ONS, 2019), it is not the main cause of the uncertainty, as this arises from misclassification of small businesses as domestic within published data. Some overestimation occurs in areas with buildings that are not typically temperature controlled (e.g. warehouses, factories) as DASH assumes all commercial spaces are temperature controlled.

Although the percentage differences in commercial annual energy consumption are larger than for domestic (Fig. 11a.ii, 11b.ii), the actual commercial values (Fig. 11d) are more spatially similar across the city than domestic values (Fig. 11c). The most spatially disparate commercial area, containing Heathrow airport (west GL, Fig. 11d), likely has undisclosed data, hence the large difference (394.7%) of 726.8 GWh. Domestic values are more spatially similar in the less densely populated suburbs, whereas areas east of the CBD are more densely populated and more spatially variable.

The annual LA (Table 1) energy fluxes have fewer data inconsistencies when the domestic and non-domestic/commercial energy consumption are combined, allowing meter classification to be ignored. DASH $Q_F$ estimates for Greater London

(5.74 W m$^{-2}$) are lower than those found using the published meter data (7.22 W m$^{-2}$), with the greatest difference in the smallest LA, City of London (DASH gives 58.02 W m$^{-2}$ and published data gives 123.48 W m$^{-2}$). The overall spatial patterns

are similar, with greater values towards the city centre and more consistent values in the surrounding suburbs.

Although address misallocation (section 3.2) is expected to cause the observed discrepancies (i.e. apparent DASH underestimation for aggregate annual values) found in the CBD, it is not possible to quantify this uncertainty. Similarly, an underestimation is expected from DASH as the meteorological input used is for one central site (Table 5) so variations (e.g.

cooler temperatures or wind effects) are unaccounted for. This could be improved by coupling DASH with a meteorological model accounting for spatial heterogeneity.

## 7  Conclusion

DASH allows anthropogenic heat fluxes to be simulated accounting for both urban form and function, using an agent-based

structure. The impact of people's behaviours at the neighbourhood scale is captured as occupants move (10 min time step), varying by day type (e.g. week day, weekend), demographics (e.g. age), location (e.g. residential, work, school), activity (e.g. cooking, recreation, travelling to school or work), socio-economic factors (e.g. appliance availability) and in response to environmental conditions (e.g. temperature related heating use). DASH includes simple transport and building energy models to allow simulation of dynamic vehicle use, occupancy, and heating/cooling demand with subsequent release of

energy to the outdoor environment through the building fabric or   ventilation.

Evaluation of DASH in Greater London for various periods in 2015 uses a top-down inventory model (GQF) and national energy consumption statistics (as cited in Table 6, R8). Overall the model generally performs well. Some of the spatial and temporal differences may be explained by data inconsistencies in the official data (e.g. privacy related, allocation of use to

office headquarters rather than place of use). Analyses with DASH allow high spatial and temporal resolution for a wide range of time periods (demonstrated here from 10 minutes to 1 year) and large spatial extent (demonstrated from output area to mega-city). The model performance evaluation addresses a wide range of these scales (e.g. 30 min spatial patterns at OA, annual at LA scale).

The expected temporal and spatial patterns are obtained (e.g. two diurnal peaks and larger fluxes in the city centre). Given DASH's capabilities these can be explored and explained. For example, domestic building $Q_{F,B}$ is more intense towards the city centre than in outer suburbs, following residential population density. The morning and evening peaks are linked to active occupancy and appliance power demand.

As DASH is demonstrated to be able to reproduce conditions generally, future work will investigate dynamic feedbacks within a city from changes in urban form and function. DASH is designed to allow parameters to be altered spatially, thus impacts on $Q_F$ emissions can be assessed. Changes may be both slow (i.e. over years) such as from an aging population, new technology uptake (e.g. change of vehicle fuels and efficiency), or governance (e.g. national energy or carbon goals) and short-term (i.e. hours, days to months) such as traffic restrictions (e.g. roadworks, flooding) changing flow. The model

performance suggests that other capabilities (e.g. additional transport types) and feedback on other variables (e.g. $CO_2$) emissions are warranted in the future. With DASH coupled to an urban land surface model, the impacts can be assessed both on $Q_F$ itself (e.g. a traffic disruption at one point in terms of the impact on $Q_{F,B}$) and its feedback on other surface energy balance terms and near-surface urban temperatures.





## 8 Acknowledgements

This work has been funded by EPSRC (doctoral training grant), NERC APEx, and Newton Fund/ Met CSSP China (SG). Dr Andy Gabey provided support in early development of DASH and adaptation Office of GQF for evaluation purposes. George Meachim carried out the gap filling for use of meteorological data.

## 9 Appendix A: Creation of Markov chains

A Markov transition matrix (Hermanns, 2003; Sericola, 2013) is built from the probabilities of transition from one state to another in the next time step, with $n$ states forming an $n \times n$ Markov transition matrix (Table A1a). Entries are the probabilities $p$ of transitioning from one state at time step $t$ (row) to another at time step $t + 1$ (column) (e.g. Tables A1b,c). Stationary distribution for state 1:

$$\pi(t) = [p(t)_{1,1}, p(t)_{1,2}, p(t)_{1,3}, p(t)_{1,4}, p(t)_{1,5}, p(t)_{1,6}] \qquad (A1)$$

The transition matrices created for this model are time inhomogeneous, reflecting a realistic diurnal profile with changes in likelihood state through the day. If state transition $n, n$ is chosen, the state does not change. Markov transition matrices may exclude entry to particular states by setting the column and row of a restricted state to zero.

As there is no way to determine the states prior to the start of a model run and to ensure no spin-up is required, the stationary distribution for the first-time step in the run is given by the diagonal of the matrix (e.g. based on Table A1 six states):

$$\pi(t) = [p(t)_{1,1}, p(t)_{2,2}, p(t)_{3,3}, p(t)_{4,4}, p(t)_{5,5}, p(t)_{6,6}] \qquad (A2)$$

This represents the distribution across states that are not in transition during the previous or the current time step.

For travel (Section 2.4.2) at $t=1$, $O_C$ are distributed using a weighted choice with the diagonal of the transition matrix (eq. A2) for that time step and age group as the weight distribution. At each subsequent time step, the origin $A_N$ has a choice to keep each $O_C$ or release them into another $a_S^N$, according to weighted choice (eq. 3) using the transition probabilities dictated by the origin $a_S^N$'s stationary distribution (eq. A1) at $t$ as $\omega$. The $A_N$ destination depends on the destination $a_S^N$ selected. If $a_S^N$ for the next time step is the same as the previous time step, the $A_N$ does not release the $O_C$.

## 10 Appendix B: Heat exchange within STEBBS

STEBBS employs a nodal approach (Foucquier et al., 2013) as found in commonly used simulation tools such as TrnSys (Klein et al., 2017) and EnergyPlus (Crawley et al., 2000). Each node represents a homogeneous component of the building, with heat transfer equations solved between each node (Figure B1). Each building component is modelled as 1-layer with bulk or effective thermal properties that account for the external and internal surfaces (e.g. a wall cavity and insulation layers are not modelled separately). As this is computationally cheap, it allows multiple instances for each $A_N$ at high temporal resolution. The only latent heat consideration is that of people from metabolic processes (Section 2.4.1).

The STEBBS considers heat exchanges by convection, conduction, and radiation, and heat gain from solar insolation and casual heat sources (Fig. B1). The convective flux, $q_{cv}$, between a fluid $f$ and a surface $s$ (Bergman et al., 2017) is:

$$q_{cv} = h\,A(T_f - T_s) \qquad (B1)$$

where $T_f$ and $T_s$ are the temperatures of the fluid ($f$) and surface ($s$), respectively, and $A$ the surface area of the building. Convective fluxes occur between indoor (outdoor) air and internal (external) wall/window/floor surface as well as the internal mass surface. For DHW, eq. B1 calculates convective flux between water and hot water tank/vessel walls. Forced convection $h$ is experienced on external walls as a function of wind speed $ws$ (m s$^{-1}$) at roof height, so is variable whilst internal values are held constant (Cole and Sturrock, 1977):




$$h = 5.8 + 4.1ws \qquad (B2)$$

Conduction between internal and external surfaces of a component (i.e. wall, window, floor, hot water tank/vessel, and ground floor to ground) is:

$$q_{cd} = k_e \, A \, \frac{T_{si} - T_{so}}{L} \qquad (B3)$$

where $k_e$ is the effective conductivity of a building component with 1 to $n$ layers of thickness $L_n$ (sum to $L$) and conductivity $k_n$:

$$k_e = \frac{L}{\frac{L_1}{k_1} + \frac{L_2}{k_2} + \cdots + \frac{L_n}{k_n}} \qquad (B4)$$

and $T_{si,}$ $T_{so}$ are the component's inside and outside surface temperatures, respectively. This is calculated for inside surfaces of a wall, ceiling, window, floor, hot water tank and hot water vessel components and their respective outside surfaces, as well as the point of contact between the ground floor and the external ground.

Shortwave insolation ($G_K$) is considered on building walls/roof and windows, with transmitted proportion through windows added to internal heat gain and absorbed proportion contributing to wall/roof/window gains (Underwood and Yik, 2004). Windows have an effective shortwave transmissivity ($\tau$) and albedo ($\Theta$), whereas walls/roof depend only on their albedo. Solar internal heat gain ($q_{si}$) as:

$$q_{si} = \tau . G_k \qquad (B5)$$

and solar gain to external wall ($q_{se}^a$) and window ($q_{se}^i$) as:

$$q_{se}^a = (1 - \Theta). G_k \ \text{ and } \ q_{se}^i = (1 - \tau - \Theta). G_k \qquad (B6)$$

The net longwave radiation ($Q_{L*}$) exchange between building surfaces (walls or windows) and surfaces (including sky) in their view is found using Bergman et al. (2017):

$$Q_{L*} = A \sum_{i=1}^n \left[ \psi_i \, \sigma \, \varepsilon \left( T_{so}^4 - T_{s,i}^4 \right) \right] \qquad (B7)$$

where $\sigma$ is the Stefan-Boltzmann constant ($5.67 \times 10^{-8}$ W m$^{-2}$ K$^{-4}$), $\varepsilon$ is the wall/window emissivity and surface temperature. $T_{s,i}$ is the temperature of the surface ($i$) in view.

The three view factors ($\psi_i$) for external wall/window surfaces (sky $\psi_s$, buildings $\psi_b$, and ground $\psi_g$) will sum to 1. Currently, for neither short nor longwave radiation are $\psi$ accounted for (i.e. uniform temperature is assumed). This could be improved when coupled with more detailed morphology data and urban meteorology as $\psi$ varies across a city with height (building facet) and density of buildings (Grimmond et al., 2001). Internal wall radiative exchanges are currently not considered.

## 11 Appendix C: Parameter values

**Table C1:** Appliances used in domestic and workplace subareas and their attributes. Usage categories: Active only (AO) consume energy as a results of user activities; Active with standby (AS) consume less when not in active use (standby); Continuous (C) have constant power consumption independent of human activity (cycling appliance power converted to continuous). See Table 3 for references.

| Appliance | Attributed activity | Usage category | Power rating (W) | Standby power rating (W) | Proportion on standby | Market permeation |
|---|---|---|---|---|---|---|
| *Domestic appliances* | | | | | | |
| Oven | Food preparation | AO | 2125 | - | - | 0.616 |
| TV | Watching TV | AS | 124 | 3 | 1 | 0.977 |
| Desktop | Computer use | AS | 100 | 20 | 1 | 0.35 |
| Laptop | Computer use | AS | 70 | 10 | 1 | 0.71 |
| Iron | Ironing | AO | 1000 | - | - | 0.9 |
| Washing machine | Laundry | AS | 792 | 1 | 0.5 | 0.93 |





| Chest fridge | - | C | 38 | - | - | 1 |
| Small appliance (generic) | - | C | 2 | - | - | |
| Lighting (single bulb) | Active | AO | 43 | - | - | - |
| *Workplace appliances* | | | | | | |
| Office "desk" | At work | AS | 250 | 25 | 0.5 | per worker |
| Office background (e.g. IT equipment) | - | C | 230 | - | - | per worker |
| Lighting | At work | AS | 120 | 120 | 0.5 | per worker |

**Table C2:** Applied building component properties to all instances of STEBBS model (Pre-1965 [House & Bungalow/Flat], Post-1965), regardless of use type and building size. *L* thickness, $\varepsilon$ emissivity, $\tau$ effective transmissivity, $\Theta$ surface albedo, $k_e$ effective thermal conductivity, $\rho$ density, $c_p$ specific heat capacity of air at constant pressure, *h* convection coefficient, $V_{FR}$ volumetric flow rate of DHW for single use domestic/commercial, $V_R$ ventilation rate, $V_T$ DHW tank volume, WWR window-to-wall ratio. Tank n = number of people per household. Vessels all other storage of DHW. For data sources refer to Table 3. * variable by wind speed, **per water user (domestic/non-domestic)

| | | $L$ (m) | $\varepsilon, \tau, \Theta$ | $k_e$ (W m$^{-1}$ K$^{-1}$) | $\rho$ (kg m$^{-3}$) | $c_p$ (J kg$^{-1}$ K$^{-1}$) | $h$ (W m$^{-2}$ K$^{-1}$) | | $V_{FR}, V_R$** (10$^{-3}$ m$^3$ s$^{-1}$) | $V_T$ (m$^3$) | WWR |
| | | | | | | | Int. | Ext. | | | |
|---|---|---|---|---|---|---|---|---|---|---|---|
| Building Fabric | External Wall/Roof | 0.241/0.327, 0.373 | 0.9,0,0.6 | 0.837/0.835, 0.104 | 1692/1690, 1076 | 803.1/804.1, 865.9 | 3 | var* | 600 | - | 0.4 |
| | Window | 0.005,0.02 | 0.88,0.9,0.05 | 1.05, 0.041 | 2500, 1000.7 | 840, 902.4 | 3 | var* | | - | |
| | Ground Floor | 0.5 | - | 0.752, 0.690 | 1540, 1470 | 1012.8, 1016 | 2.8 | - | | - | - |
| | Internal Mass | - | 0.91, 0, 0 | 0.121 | 873.7 | 967.9 | 3 | - | | - | - |
| DHW | Tank n = 1 | 0.055 | 0.9, -, - | 0.0275 | 745.55 | 1380 | 243 | 3 | 0.183 / 0.15 | 0.115 | - |
| | Tank n = 2 | | | | | | | | | 0.115 | |
| | Tank n = 3 | | | | | | | | | 0.125 | |
| | Tank n = 4 | | | | | | | | | 0.148 | |
| | Tank n = 5 | | | | | | | | | 0.170 | |
| | Tank n = 6 | | | | | | | | | 0.180 | |
| | Vessels | 0.0047 | 0.91, -, - | 0.16 | 1380 | 1380 | 243 | 3 | 0.1372 / 0.1125 | - | - |
| External Ground | | 2 | - | 1.28 | - | - | - | - | - | - | - |
| Internal Air | | - | - | - | - | 1005 | - | - | - | - | - |

## 12 Appendix D: Code availability/Data availability

All code and data are deposited at 10.5281/zenodo.3745524. The archive will be made available once the paper is accepted.



**Table D1:** Data examples. More details (example structure, units, raw data source, location in repository and location of use in code) can be found at: https://github.com/Urban-Meteorology-Reading/DASH-X/wiki/All-data

| | Filename | File type | Definition |
|---|---|---|---|
| ***i) Population*** | | | |
| a | age_groups | csv | Population of each age group in each $A_N$ |
| b | allworkers | csv | Residential and workplace spatial unit relation |
| c | area_hierarchy | csv | List of $A_N$ in the larger, containing, spatial unit (B) |
| d | daytype | csv | Dates used by run and corresponding day of year and day type |
| e | SchoolWorkShopcap | csv | School and workplace populations and shops and 'other' subarea capacities for each $A_N$ |
| ***ii) Transport*** | | | |
| a | SpatialUnitRoadLengths | csv | $L_r$ in each B |
| b | average_passengers | csv | Average number of people in a single $m$ vehicle |
| c | distance_freqs | csv | Journey distance categories and their respective mode weightings |
| d | fuel_consumption | csv | Average urban fuel consumption for urban roads for vehicle stock (g km$^{-1}$) |
| e | fuel_ratio | csv | Proportions of each $m$ using each $f$ |
| f | IntraBorDist/$x$matrix | csv | Distance matrix for distance between $A_N$ centroids in B |
| g | IndivBor/$x$h_wsorted | csv | Proportions of people using each mode to travel from home to work |
| h | IndivBor/$x$w_hsorted | csv | Proportions of people using each mode to travel from work to home |
| i | MeanSpeedLimits | csv | Mean $v_{r,lim}$ for each $r$ in each B |
| j | RoadAADTMeansLengthWeighted | csv | AADT means of each $r$, mode for each B |
| k | routes_distances | csv | List of route segment distances for each spatial unit traversed for each route |
| l | routes_int | csv | List of routes between each start-destination pair, including the spatial units traversed for each route |
| m | route_reference_matrix | csv | Reference matrix for route numbers |
| n | ShopsGravity | csv | Gravity weightings (Eq. 3) for travel to shops and other subareas |
| o | speed_fuel_ratio_func | pickle | Functions of normalised speed - fuel consumption relation for each $m$ |
| p | traveltime_functions | pickle | Functions relating distance to time travelled for each mode |
| q | vehicle_length (in settings.nml) | - | Length of representative vehicle |
| ***iii) Area*** | | | |
| a | env_vars | csv | Environmental variables used for each time step |
| b | IndustrialOAs | csv | Location of industrial land use around the study area |
| c | OA_area_details | csv | Population, road length, building stock and dimensions, floor plan area data for each $A_N$ |
| ***iv) Buildings*** | | | |
| a | CommBuildingArchetype | nml | Multiple *.nml* lists for each commercial building archetype and their STEBBS parameters |
| b | CommTypes | nml | Multiple *.nml* lists for each school/shops/other land use type and their parameters |
| c | DomApplianceList | nml | Multiple *.nml* lists for appliances used by occupants in domestic buildings, and their parameters |
| d | DomBuildingArchetype | nml | Multiple *.nml* lists for each commercial building archetype and their STEBBS parameters |
| e | domlighting | nml | Parameters for domestic lighting |
| f | WorkApplianceList | nml | Multiple *.nml* lists for appliances used by occupants in commercial buildings, and their parameters |
| g | $x$personactiveweekend/day | csv | Proportions of people active (awake and present) in households with $x$ people present at each time step |
| h | $x$personweekend/day | csv | Proportions of people who belong to household of size $x$ present in household at each time step, given that someone is present |



Table D.2 Archive of model runs (Table 6) inputs and results

| R | Zenodo reference | Extent | Area | Spatial Scale | Temporal Scale |
|---|---|---|---|---|---|
| 1 | 10.5281/zenodo.3745524 | GL | Camden | $a_N^x$ | 10 min |
| 2 | 10.5281/zenodo.3745524 | GL | GL | $A_N$ | 10 min |
| 3 | 10.5281/zenodo.3745524 | GL | E00023911, E00015661, E00008490 | $A_N$ | 10 min |
| 4 | 10.5281/zenodo.3745524 | GL | GL | LA | 10 min |
| 5 | 10.5281/zenodo.3745524 | GL | GL | GL, LA | 10 min |
| 6 | 10.5281/zenodo.3745524 | GL | GL | OA ($A_N$) | As Run 5 |
| 7 | 10.5281/zenodo.3745524 | GL | GL | GL | Annual |
| 8 | 10.5281/zenodo.3745524 | GL | GL | LSOA - dom, MSOA - n-dom | Annual |

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





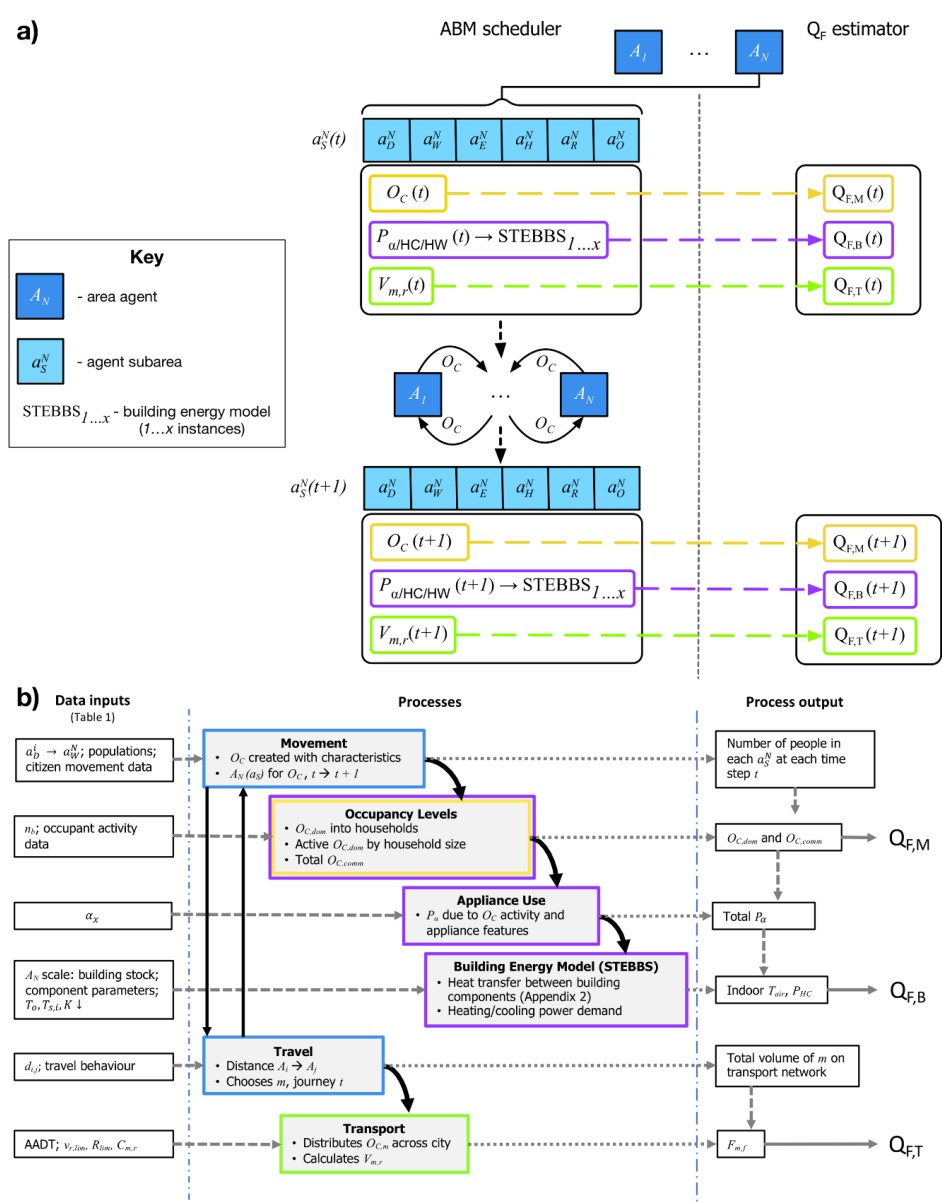

**Figure 1:** Overview of DASH (**a**) Agent-agent interaction and estimation of $Q_F$ with the $A_N$ (mid blue) to $a_S^N$ (light blue) relations, changes in process outputs (yellow, purple, green) between time steps and the reaction (arrows) to give $Q_F$. (**b**) Processes include agent-agent interactions (blue boxes), agent reaction and interaction with environment ($Q_{F,B}$: purple, $Q_{F,M}$: yellow, $Q_{F,T}$: green boxes), inputs (dashed lines), process outputs (dotted lines) and their interactions (thick lines), and $Q_F$ outputs (solid grey lines). Notation list gives definitions.






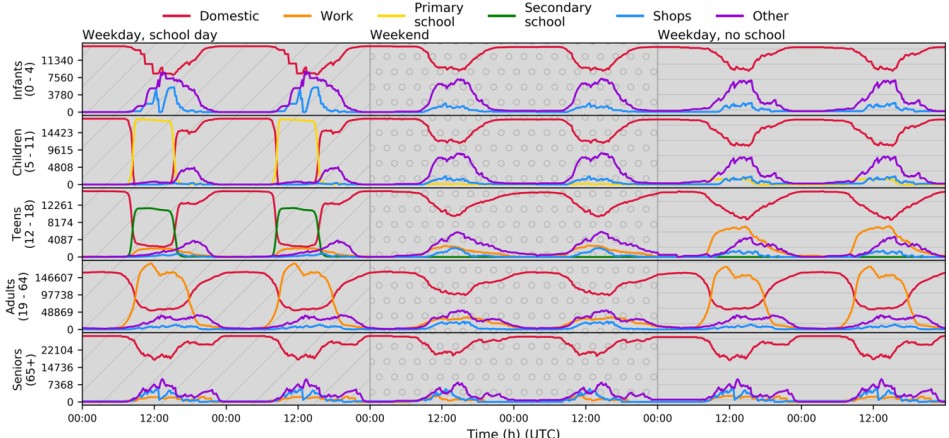

**Figure 2:** Total occupancy of each $a_s^N$ in one LA for five age groups across six consecutive days of three types (textured background): SW (diagonal lines); WE (dotted); NS (horizontal lines) Run 1 (Table 6).

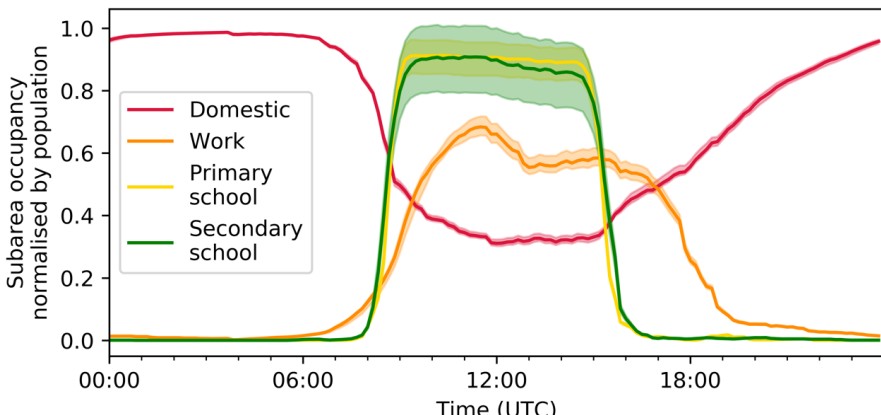

**Figure 3:** Median (line) and IQR (shading) of total occupancy of each $a_S^N$ in Greater London for one weekday (R2, Table 6), normalised by actual static population (Table 2).

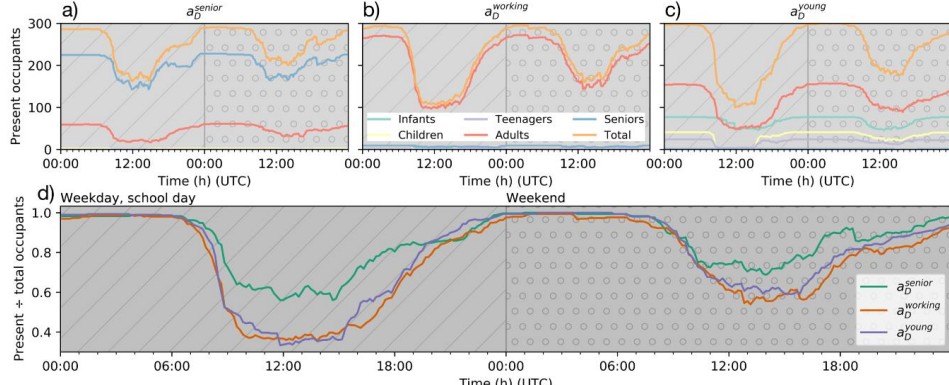

**Figure 4:** Present occupancy levels (R3, Table 6) in three $a_D^N$ by day type (textured background) **a)** $a_D^{senior}$ (number of people per age group living in the area: 0 infants, 2 children, 0 teenagers, 61 adults, 228 seniors); **b)** $a_D^{working}$ (5 infants, 6 children, 3 teenagers, 274 adults, 9 seniors); **c)** $a_D^{young}$ (77 infants, 41 children, 24 teenagers, 157 adults, 1 senior). **d)** Normalised total occupancy levels for the three $a_D^N$.



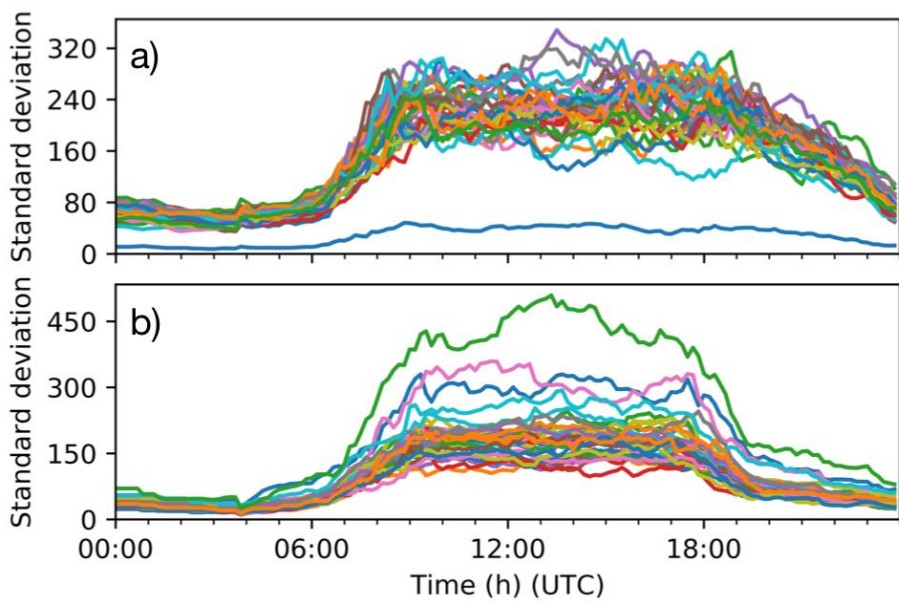

**Figure 5:** Standard deviation of LA (all boroughs of London, colour, for 44 weekdays (preceded by weekdays)) occupancy levels (R4, Table 6)) for: (**a**) $a_D^N$ and (**b**) $a_W^N$.

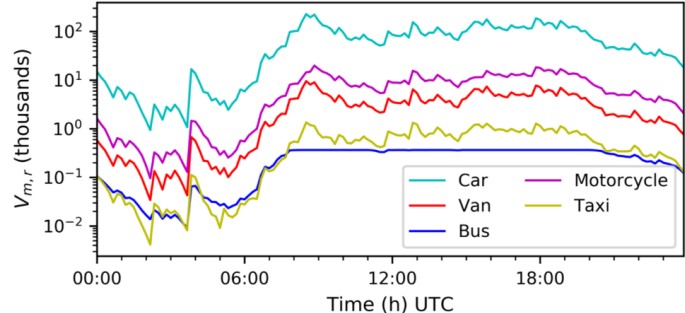


**Figure 6:** Simulated volume of vehicles across Greater London for 19 June 2015 (R5, Table 6).

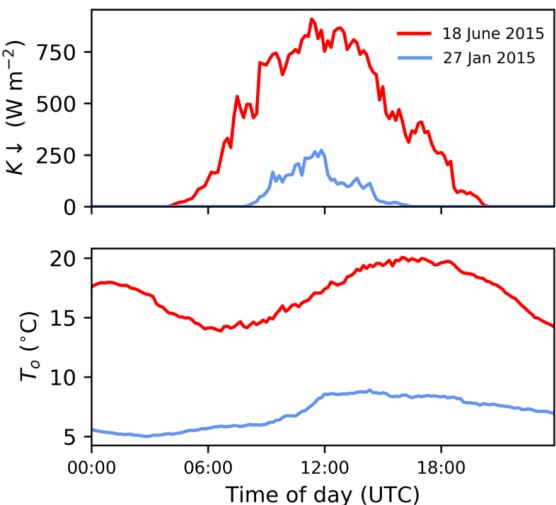

**Figure 7:** Incoming shortwave radiation ($K{\downarrow}$, W m$^{-2}$) and outdoor air temperature ($T_o$, °C) for two SW days. Observations (Table 5) are assumed to be constant across the domain in all runs (Table 6).

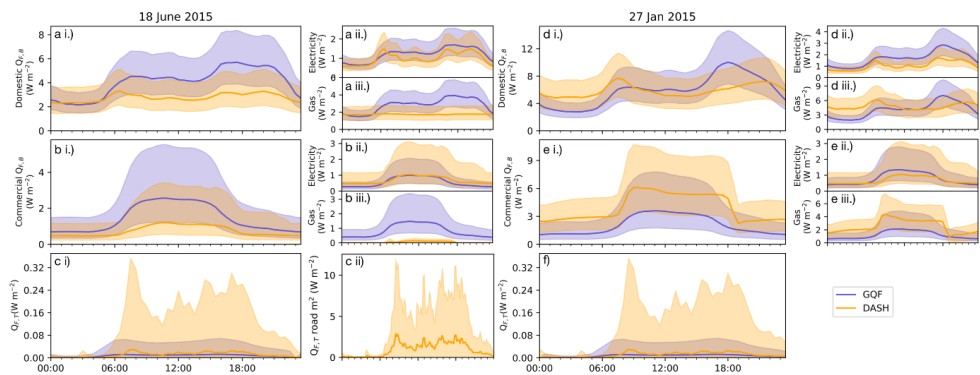

**Figure 8:** Analysis of $Q_F$ (R6, Table 6) median (line) and IQR (shading) for two days in 2015: (**a, b, c**) 18 June and (**d, e, f**)
27 January; showing total $Q_{F,B}$ for (**a.i, d.i**) domestic, (**b.i, e.i**) commercial; with (**a.ii, d.ii**) domestic electricity (GQF) or appliance power demand (DASH); (**a.iii, d.iii**) domestic gas (GQF) or heating + cooling + hot water demand (DASH); (**b.ii, e.ii**) commercial electricity (GQF) or appliance power demand (DASH); (**b.iii, e.iii**) commercial gas (GQF) or heating + cooling + hot water demand (DASH); and (**c, f**) $Q_{F,T}$ at $A_N$ scale; and (**c.ii**) $Q_{F,T}$ for road area only. Fig. 7 shows weather conditions. Fig. 9 shows absolute errors between the two models.





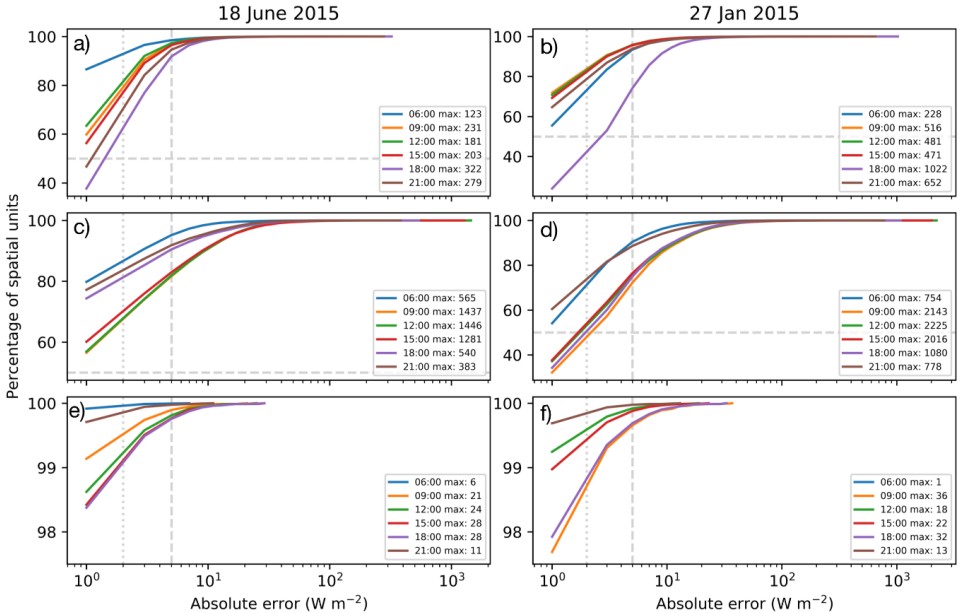


**Figure 9:** Ranked cumulative frequency of spatial $AE_i$ (eq. 20b) with 2, 5 W m$^{-2}$ (vertical lines) and maximum (key, W m$^{-2}$) indicated at six times (colour) for R6 (Table 6, Fig. 8) on two days in 2015: (**a, c, e**) 18 June 2015, (**b, d, f**) 27 January 2015, for (**a, b**) total domestic $Q_{F,B}$, (**c, d**) total commercial $Q_{F,B}$, and (**e, f**) total transport $Q_{F,T}$. Note y-axes are different between rows (50 % of spatial units shown by horizontal dashed line if applicable) and x-axes are log$_{10}$.


**Figure 10:** Daily (1 October 2014 - 30 September 2015) DASH normalised total heating/cooling and domestic hot water (DHW) energy demand (R7, Table 6) for Greater London, minimum and maximum London outdoor air temperature (°C) (Table 5) and normalised national gas demand (NG, 2015). See section 3.2 for normalisation.




**Figure 11:** DASH (R8, Table 6) $nE_i$ of total energy consumption represented by (**i**) choropleth and (**ii**) histogram for (**a**) LSOA scale domestic use and (**b**) MSOA scale commercial use. $AnE_i$ of total energy consumption for (**c**) LSOA scale domestic and (**d**) MSOA scale commercial. Annual average energy consumption at LA scale for (**e**) reference data and (**f**) DASH.





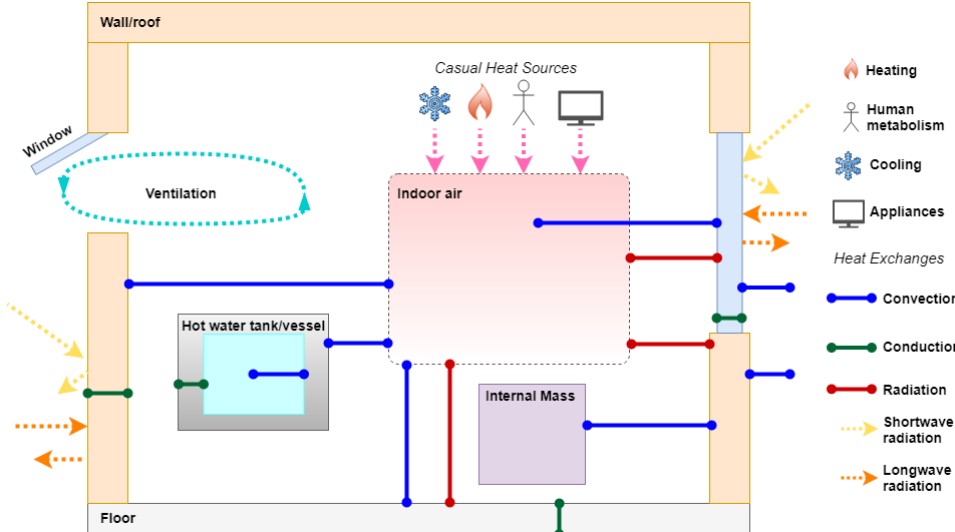

**Figure B1:** STEBBS 1-D model simulates building facets/nodes (dots), casual heat sources and heat exchanges. Longwave radiation is absorbed by building facets from the outdoor environment, and shortwave radiation from direct, diffuse and reflected sources.

**Table 1:** Sources of data used by DASH and the highest spatial resolution (columns) used in Greater London. Details are given in the other Tables (Tab) and Appendices (App) indicated. Notation defined in text.

| Data Category | Spatial Scale | $A_N$ | | | B | | London/National |
|---|---|---|---|---|---|---|---|
| | | OA | LSOA | MSOA | LA | City | |
| Population | | Tab 2 | | | | | |
| Activities | | | | | | App. A | |
| Appliance | | | | | | Tab C1 | |
| Building | Size | Tab 4 | | | | | |
| | Types | | | | | Tab 4 | |
| | Properties | | | | | Tab 4 | Tab C2 |
| Trans-port | Mode Attributes | | | | | | Tab 3 |
| | Route speed limits | | | | Tab 3 | | |
| | Mode & route capacity | Tab 3 | | | | | |
| Environmental conditions | | | | | | Tab 5 | |





**Table 2:** Spatial, temporal, and demographic data used to inform activity in Greater London. *Data sources*: Greater London Authority (GLA), Office for National Statistics (ONS), Chartered Institution of Building Service Engineers (CIBSE), Ordnance Survey (OS), Valuation Office Agency (VOA). See also Table D1.

| Data Category | Model Application | Data Source |
|---|---|---|
| Area Codes | all $B$ – LSOA, MSOA, LA<br>all $A_N$ – OA | GLA (2011) |
| Centroid | all $B$, $A_N$ | GLA (2011) - GIS shapefiles |
| Area | $A_N$ - OA | |
| Population | Domestic ($a_D^N$) - # by age cohort [all] | ONS (2015) |
| | Workplace ($a_W^N$) - # by age cohort [Teen/Adult/Senior] | ONS (2014a) |
| | Primary school ($a_E^N$) - # registered [Child] | GLA (2014) |
| | Secondary School ($a_H^N$) - # registered [Teen] | |
| | Shops ($a_R^N$) - # of shops | OpenStreetMap (2017) |
| | Other ($a_O^N$) - # of businesses | |
| Household | Domestic ($a_D^N$) – distribution by # of $O_C$ per house | ONS (2011) |
| Age cohort | Infant [0-4 years] - # in $a_D^N$ | ONS (2015) |
| | Child [5-11 years] - # in $a_D^N$, $a_E^N$ | |
| | Teen [12-18 years] - # in $a_D^N$, $a_H^N$, $a_W^N$ | |
| | Adult [19-64 years] - # in $a_D^N$, $a_W^N$ | |
| | Seniors [65+ years] - # in $a_D^N$, $a_W^N$ | |
| Anchor locations | $A_N$ - # of residence/workers/students as function of age | - |
| Day Types (to inform activity profiling) | School weekday [by age cohorts: Child/Teen/Adult] | Gershuny and Sullivan (2017) |
| | Weekend [by all age cohorts] | |
| | Public holiday [by all age cohorts – as weekend] | |
| | Non-school weekday [by age cohorts: Child/Teen/Adult and # of dependent children in different households] | ONS (2017a), Gershuny and Sullivan (2017) |
| Initiation of travel | Clock time of start of journeys within city and subareas | Gershuny and Sullivan (2017) |
| Building Archetypes assigned Areas | Typical height (m), depth (m) and total floor area (m) of identified types. Height: depth ratios: House 9:12.5, Bungalow 5.5:12.5, Low-rise flats: 6.1:20. Width calculated to maintain ratio and total building volume. | VOA (2015), Butcher and Craig (2016), Mavrogianni et al. (2012) |
| | Floor Plan Area (m$^2$) and average height (m) to give volume. | OS (2014) |


**Table 3:** Data sources for physical building characteristics applied to building archetypes. Symbols notation table. Symbols used are: $L$ wall thickness (m), $\rho$ building material density (kg m$^{-3}$), $k_e$ wall effective thermal conductivity (W m$^{-1}$ K$^{-1}$), $\varepsilon$ emissivity, $h$ convection coefficient (W m$^{-2}$ K$^{-1}$), $V_T$ (m$^3$) volume of tank (dependent on number of persons per household), **ToU** time of use. **Data Sources:** §$_1$ – British Council for Offices (BCO (2009)), §$_2$ - Richardson et al. (2010), §$_3$ - DECC and BRE (2016), §$_4$ - Hawkins (2011), §$_5$ - DECC (2015), §$_6$ - HCA (2010), §$_7$ - Butcher (2004). §$_2$ used for cycling patterns of continuously on appliances (i.e. fridge/freezer). See also Table D1.


| Characteristic | | | Domestic | Non-Domestic |
|---|---|---|---|---|
| Building dimension | Height / Floor Plan | | Mavrogianni et al. (2012), OS (2014) | |
| | WWR | | Butcher (2012) | |
| Thermophysical properties (Table C2) | Building | $L$, $\rho$, $k_e$ | Butcher and Craig (2016) | |
| | | $\varepsilon$, $c_p$ | Stewart et al. (2014), Butcher and Craig (2016) | |
| | | $V_R$ | Butcher (2014) | |
| | | Internal $h$ | Butcher and Craig (2016) | |
| | | External $h$ | Cole and Sturrock (1977) | |
| | External | $k_{ground}$ | Butcher and Craig (2016) | |
| DHW Services (Tank/Pipes) | $L$ | | Flamco (2017) | |
| | $\varepsilon$, $c_p$, $k_e$ | | Butcher and Craig (2016), Flamco (2017) | |
| | $\rho$ | | Butcher and Craig (2016), Flamco (2017) | |
| | $h$ | | Butcher and Craig (2016), Knudsen (2002) | |
| | $V_T$ | | MWS (2019) | IOP (2002) |
| Power Ratings (W) (*Table C1) | Heating/ Cooling | | Butcher and Craig (2016) | |
| | DHW | | Flamco (2017), Palmer (2016) | |
| | Appliance* | | §$_1$ §$_2$ §$_3$ §$_7$ | §$_4$ §$_5$ §$_6$ |
| Activity | Appliance $\alpha_{j,k}$ | | §$_7$, BCO (2009), DECC and BRE (2016) | |
| | DHW $V_{FR}$ | | BSI (1997), Butcher (2014) | |
| | ToU | | Gershuny and Sullivan (2017) | |



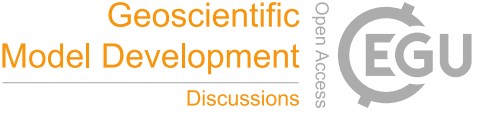

**Table 4:** Transport mode and route attributes used in model. Fixed, homogeneous, values given explicitly. Note that buses assumed all diesel (85% of fleet in 2015, rest mostly 'Hybrid') and other electric and low emission vehicles are excluded (plugin cars reported as 0.2% of registered vehicles in GL in 2015 from DfT and DVLA (2019)). **Data sources:** $\S_8$ ONS (2014b), $\S_9$ ONS (2018), $\S_{10}$ DfT (2017), $\S_{11}$ DfT (2014a, 2014b), $\S_{12}$ London Datastore (2014), $\S_{13}$ OS (2016), $\S_{14}$ Smith et al. (2009), $\S_{15}$ Highways Agency (2017), $\S_{16}$ TfL (2018), $\S_{17}$ TfL (2019), $\S_{18}$ OS (2015), $\S_{19}$ TfL Train and Underground Rolling Stock Information Sheets from $\S_{10}$, $\S_{20}$ TfL working timetables from $\S_{10}$, $\S_{21}$ Iamarino et al. (2012). Annual Average Daily Traffic AADT.

| Mode (m) | People per vehicle ($\S_{10}$, $\S_{15}$) | Fuel Use Ratio (petrol: diesel: electric) $\S_{10}$, $\S_{17}$ | $Q_{F,M}$ per passenger (W m⁻²) $\S_{21}$ ᐃNot applied in evaluation | Heat emission of ^ combustion and by * speed & fuel (*electric not considered*) | Commute mode Choice ($a_D^i \to a_W^i$) | Journey Time | Constraints/Limits — Proportion of mode by route | (Mode, Route) capacity ($C_{m,r}$, $R_{lim}$) and [speed] [$v_{r,lim}$] limit | Route ($r$) Dimensions (e.g. length, # of lanes, # of tracks) ᐃNot applied in evaluation |
|---|---|---|---|---|---|---|---|---|---|
| Car | 1.4 | 0.84:0.16:0 | 70 | ^ $\S_9$ | $\S_8$ | $\S_{10}$ | $\S_{11}$ | AADT – $\S_{12}$ | $\S_{12}$, $\S_{13}$, $\S_{16}$ |
| Van | 1.4 | 0.1:0.9:0 | 70 | * $\S_{14}$ | | | | Speed – $\S_{18}$ | |
| Taxi | 2.5 | 0:1:0 | 70 | | | | | | |
| Motorcycle | 1 | 1:0:0 | 70 | | | | | | |
| Bus/coach | 17.3 | 0:1:0 | 55 | | | | | | |
| Metro | -- | -- | 62 ᐃ | -- | | | -- | $\S_{19}$, $\S_{20}$ ᐃ | $\S_{19}$ ᐃ |
| Surface rail | -- | -- | 55 ᐃ | -- | | | -- | | |
| Bicycle | 1 | -- | 230 | -- | | | -- | -- | -- |
| Walking | 1 | -- | 140 | -- | | | -- | -- | -- |


**Table 5:** Observed meteorological variables at 60.9 m above ground level King's College London KSSW site (Kotthaus and Grimmond 2014, Ward et al. 2016). See Figure 1a in Kotthaus and Grimmond (2014) for site location. From these other variables are derived.

| | Meteorological Variable | Sensor |
|---|---|---|
| $T_o$ | Outdoor air temperature (°C) | Vaisala WXT 520 |
| $ws$ | Wind speed (m s⁻¹) | |
| $K_\downarrow$ | Incoming shortwave radiation (W m⁻²) | Kipp & Zonen CNR4 Net Radiometer |
| $Q_{L\uparrow}$ | Outgoing longwave radiation (W m⁻²) | |


**Table 6:** DASH model runs (R) setup. Runs are characterised by period (dates, and day types: WD weekdays), areal extent (Table 1, dom: domestic, n-dom: non-domestic). Data sources: $\S_{22}$ GLA (2014), $\S_{23}$ ONS (2015), $\S_{24}$ ONS (2014a), $\S_{25}$ National Grid (NG, 2015), $\S_{26}$ BEIS (2017c), $\S_{26}$ BEIS (2017c). Other details are given in Table D.2 and Section 2.

| R | Period | Extent run | Area Analysed | Spatial Scale | Spin-up (days) | Evaluation Data | Temporal Scale | Fig. |
|---|---|---|---|---|---|---|---|---|
| 1 | 12 – 17 Feb 2015 | GL | Camden | $a_N^x$ | - | - | 10 min | 2 |
| 2 | 12 Feb 2015 | GL | GL | $A_N$ | - | $\S_{22}$, $\S_{23}$, $\S_{24}$ | 10 min | 3 |
| 3 | 13 – 14 Feb 2015 | GL | E00023911, E00015661, E00008490 | $A_N$ | - | - | 10 min | 4 |
| 4 | First 44 WD of 2015 preceded by WD 6-9, 13-16, 20-23, 27-30 Jan, 3-6, 10-13, 24-27 Feb, 3-6, 10-13, 17-20, 24-27 Mar 2015 | GL | GL | LA | - | - | 10 min | 5 |
| 5 | 19 June 2015 | GL | GL | GL, LA | 1 | - | 10 min | 6 |
| 6 | 19 June 2015, 27 Jan 2015 | GL | GL | OA ($A_N$) | 1 | GQF | 30 min | 8, 9 |
| 7 | 1 Oct 2014 – 30 Sept 2015 | GL | GL | GL | 1 | $\S_{25}$ | Annual | 10 |
| 8 | 1 Oct 2014 – 30 Sept 2015 | GL | GL | LSOA - dom, MSOA - n-dom | 1 | $\S_{26}$ | Annual | 11 |





**Table A1:** Markov transition matrix (**a**) general for six states (rows and columns) (**b**) data for a single time step and (**c**) transition probabilities for the data in (b) (Gershuny and Sullivan, 2017)

| a) | 1 | 2 | 3 | 4 | 5 | 6 |
|---|---|---|---|---|---|---|
| 1 | $p(t)_{1,1}$ | $p(t)_{1,2}$ | $p(t)_{1,3}$ | $p(t)_{1,4}$ | $p(t)_{1,5}$ | $p(t)_{1,6}$ |
| 2 | $p(t)_{2,1}$ | $p(t)_{2,2}$ | $p(t)_{2,3}$ | $p(t)_{2,4}$ | $p(t)_{2,5}$ | $p(t)_{2,6}$ |
| 3 | $p(t)_{3,1}$ | $p(t)_{3,2}$ | $p(t)_{3,3}$ | $p(t)_{3,4}$ | $p(t)_{3,5}$ | $p(t)_{3,6}$ |
| 4 | $p(t)_{4,1}$ | $p(t)_{4,2}$ | $p(t)_{4,3}$ | $p(t)_{4,4}$ | $p(t)_{4,5}$ | $p(t)_{4,6}$ |
| 5 | $p(t)_{5,1}$ | $p(t)_{5,2}$ | $p(t)_{5,3}$ | $p(t)_{5,4}$ | $p(t)_{5,5}$ | $p(t)_{5,6}$ |
| 6 | $p(t)_{6,1}$ | $p(t)_{6,2}$ | $p(t)_{6,3}$ | $p(t)_{6,4}$ | $p(t)_{6,5}$ | $p(t)_{6,6}$ |

| b) | Domestic | Workplace | Shops | Other |
|---|---|---|---|---|
| Domestic | 270 | 46 | 2 | 4 |
| Workplace | 1 | 170 | 0 | 1 |
| Shops | 0 | 0 | 5 | 0 |
| Other | 0 | 1 | 1 | 18 |

| c) | Domestic | Workplace | Shops | Other |
|---|---|---|---|---|
| Domestic | 270/320 | 46/320 | 2/320 | 4/320 |
| Workplace | 1/172 | 170/172 | 0 | 1/172 |
| Shops | 0 | 0 | 1 | 0 |
| Other | 0 | 1/20 | 1/20 | 18/20 |