# Peer review of "Isabella Capel-Timms1,2, Stefán Thor Smith2, Ting Sun1, Sue Grimmond1"

_Geoscientific Model Development, 2020_

## Short Comment (SC1) · 18 May 2020

This is an executive editor comment highlighting the ways in which this manuscript is not currently compliant with GMD policy on code and data availability.

In this case, there is a critical issue with the code and data availability. The model code and results are archived to Zenodo, which is best practice, however the Zenodo archive is restricted, and the manuscript indicates that this is pending acceptance. This sort of embargo is completely incompatible with GMD's policy on code and data availability. From the code and data availability policy:

Code and data access must be provided at the time that the discussion paper is submitted. Embargoes, whether pending acceptance or for a defined period, are not acceptable.[1]

Having the code and data embargoed undermines the public peer review process of GMD. The embargo must therefore be lifted immediately, or the manuscript will need to be rejected.

At a more technical level, placing a bare DOI in the text of the paper is not best practice. Instead, the full data citation should appear in the paper bibliography, and this should be cited from the text. Zenodo make this particularly easy by providing the correct bibliographic entries in a variety of formats at the bottom right of the archive web page.

Further details on code and data availability requirements are in the GMD model code and data policy: https://www.geoscientific-model-development.net/about/code_and_data_policy.html. The reasons for the policy and more detail are provided in this editorial: https://doi.org/10.5194/gmd-12-2215-2019.
* * ** * *
[1]https://www.geoscientific-model-development.net/about/code_and_data_policy.html

---

## Referee Comment (RC1) · Anonymous Referee #1 · 24 May 2020

Isabella et al developed the DASH to quantitatively evaluate the anthropogenic activities impacting heat emissions. This model highlighted the Spatiotemporal distribution anthropogenic heat emissions. This is an important model that have board implications to the urban area. This topic of this paper fits the scope of GMD. However, I have some concernsïijŽ

45. Please check the reference format. " (Oke, 198)"

215. It seems the authors does not consider the heat emissions generated by the metabolism during sleep, which is incomplete for this part.

410. I agree with the author's statement that "The limited consideration of building

material thermophysical properties is expected to reduce the spatial variance in heating and cooling contributions". But the dimension of the building is equally important. In the case of London, it does not take differences in building dimensions into consideration, which would be an important reason to cause the largest discrepancy in CBD except for misallocation in the published data, as the author mentioned in 635

---

## Referee Comment (RC2) · Anonymous Referee #2 · 17 Jun 2020

GENERAL COMMENTS: Overall, this manuscript presents an intriguing and thoroughly reasoned agent-based framework for evaluating the dynamically and spatially varying anthropogenic heat emissions across cities. Such a framework is superior in theory to prior approaches to estimate Qf and should be published. Nevertheless, this reviewer has several high-level concerns about the practical implementation and evaluation of such a complex model. It is likely that the authors can suitably address these issues through additions to the manuscript text, including appropriate caveats (or further explanation) regarding model accuracy.

1. At a fundamental level, this model is extremely complex with so many degrees of

freedom, and input variables/assumptions that are highly uncertain, that in practice, the model may not be any more accurate than much simpler inventory-based estimates. The authors need to make a stronger case that the added complexity increases accuracy and makes a meaningful and important difference in the anthropogenic heat profiles, and in secondary results related to the use of these profiles (e.g., estimates of the local diurnal warming signal when Qf is incorporated into atmospheric models).

2. Related to the above point, validation is foundational to determining the usefulness of a framework such as this. However, as recognized by the authors, validation is not really possible given the significant limitations of other methods of estimating Qf. Nevertheless, when the authors do compare estimates of energy consumption to actual observations (from utility data) their model does not appear to perform very well. So, any estimates of anthropogenic heating derived from the energy use estimates may be suspect.

SPECIFIC COMMENTS: 1. Lines 40-44 – This may seem like a minor point, but while eqn 1 is a commonly used representation of the energy balance for cities, it is not clearly articulated whether this is truly a surface energy balance or a volumetric energy balance. If the former, then storage is zero and Qf is minimal as most Qf is emitted directly into the air volume. If the latter, then advection would seem to be of significance in a heterogeneous urban setting. 2. Section 2.4.3 – Does STEBBS allow for a dynamic setpoint temperature? Most commercial and many residential buildings have setpoints that vary based on management (either BMS or by individual occupants). 3. Section 3 – are light manufacturing and industrial buildings taken into account in either DASH or GQF? These can be significant energy users in certain areas of larger cities, and might be ignored, potentially explaining part of the underestimation of energy use in the CBD. For that matter, can the authors provide more clarity on how many archetype buildings they use in their modeling? Some prior studies doing similar tasks have suggested 20-30 archetypes are needed to adequately capture citywide variations in residential + commercial energy consumption. 4. In section 4 (and elsewhere) can authors clarify

whether "occupancy" differentiates indoor and outdoor location of individuals? If AC is present, the rejected heat from indoor metabolism includes metabolism + AC energy used to reject it. 5. Lines 631-637 – the errors in estimates of domestic energy consumption seem rather large in general. It would seem that if the building sector archetypes are reasonable, the errors in estimating energy consumption should be much smaller. 6. What are the units in Figure 11 e and f? 7. Appendix B – STEBBS: the description of STEBBS suggests that walls and roofs are modeled as single layers with bulk properties. Can the authors clarify whether the conduction equation is solved within these single-layer constructions? If not, they would essentially be assuming steady state conduction through the wall and roof at all times–which would introduce significant errors. Other assumptions in STEBBS may be questionable, as well. I am not familiar with STEBBS. Some more details on this model would be helpful. Additionally, there are standard test cases (from ASHRAE) against which building science models are evaluated and validated. Can the authors present some quantitative data to confirm that the STEBBS model produces accurate estimates of building energy consumption for any specific building (e.g., compare model results with those of a trusted and ASHRAE-validated model such as TRNSYS or EnergyPlus for each of your archetypes). If only ~20 archetypes are simulated for a single year, this task is actually rather quick in existing software such as EnergyPlus (or DoE-2) which has already undergone extensive validation and improvement over the past 40+ years. If STEBBS is new (and less validated), can the authors make the case for why they didn't simply go with an existing model?

---

## Author Comment (AC2) · 7 Jul 2020

Thank you for your comments. The line numbers in our response referred to the updated manuscript unless otherwise stated.

**Anonymous referee #1**

Isabella et al developed the DASH to quantitatively evaluate the anthropogenic activities impacting heat emissions. This model highlighted the Spatiotemporal distribution anthropogenic heat emissions. This is an important model that have board implications to the urban area. This topic of this paper fits the scope of GMD. However, I have some concerns

45. Please check the reference format. " (Oke, 198)"
Citation has been corrected to (Oke, 1988)

215. It seems the authors does not consider the heat emissions generated by the metabolism during sleep, which is incomplete for this part.
This is accounted for. The following text has been changed to make this clearer:

**L217**:

> *Both $\beta$ and M can vary with activity (e.g. office work/sitting, walking, sleeping)*

410. I agree with the author's statement that "The limited consideration of building material thermophysical properties is expected to reduce the spatial variance in heating and cooling contributions". But the dimension of the building is equally important. In the case of London, it does not take differences in building dimensions into consideration, which would be an important reason to cause the largest discrepancy in CBD except for misallocation in the published data, as the author mentioned in 635

Total building volume is (approximately) correct for each OA (the exact distribution between archetypes is not known). One non-domestic and three domestic building types (bungalow, flat, house) are used to create the volume. The building level of detail (LoD) can vary with available data.

**L416:**

> *The limited consideration of building material thermophysical properties and dimensions is expected to reduce the spatial variance in heating and cooling contributions to $Q_F$ in DASH. DASH can use more building features given suitable input data.*

**Anonymous referee #2**

GENERAL COMMENTS: Overall, this manuscript presents an intriguing and thoroughly reasoned agent-based framework for evaluating the dynamically and spatially varying anthropogenic heat emissions across cities. Such a framework is superior in theory to prior approaches to estimate Qf and should be published. Nevertheless, this reviewer has several high-level concerns about the practical implementation and evaluation of such a complex model. It is likely that the authors can suitably address these issues through additions to the manuscript text, including appropriate caveats (or further explanation) regarding model accuracy.

1. At a fundamental level, this model is extremely complex with so many degrees of freedom, and input variables/assumptions that are highly uncertain, that in practice, the

model may not be any more accurate than much simpler inventory-based estimates. The authors need to make a stronger case that the added complexity increases accuracy and makes a meaningful and important difference in the anthropogenic heat profiles, and in secondary results related to the use of these profiles (e.g., estimates of the local diurnal warming signal when Qf is incorporated into atmospheric models).

Our goal in developing DASH is to provide a model that allows behaviour dynamics to be captured (i.e. response to localised events having larger spatial and temporal implications). This then enables the possibility of feedback. The improvements (possibly accuracy) is from not assuming that the response is static.

We agree that the model is complex with many degrees-of-freedom. An understanding of the influence of uncertainty in variables and assumptions is the subject of sensitivity analysis that is beyond the scope of this paper. Here we introduce the conceptual framing of the model, its implementation, and an evaluation for the case of London against currently accepted models.

2. Related to the above point, validation is foundational to determining the usefulness of a framework such as this. However, as recognized by the authors, validation is not really possible given the significant limitations of other methods of estimating Qf. Nevertheless, when the authors do compare estimates of energy consumption to actual observations (from utility data) their model does not appear to perform very well. So, any estimates of anthropogenic heating derived from the energy use estimates may be suspect.

As a socio-technical-physical model, the idea of "validation" is not something we aim to achieve. The evaluation carried out informs us of the similarities (differences) in spatiotemporal profiles and gives some insight into the modelling assumptions and simplifications that have been documented. Whilst differences with existing models and data are observed, these are relatively small in most cases/locations when the absolute error (e.g. Fig. 9) is considered. Despite these documented discrepancies the larger spatial patterns are captured well, with greater variance in spatial differences observed at higher resolution (Fig. 11).

SPECIFIC COMMENTS:
1. Lines 40-44 – This may seem like a minor point, but while eqn 1 is a commonly used representation of the energy balance for cities, it is not clearly articulated whether this is truly a surface energy balance or a volumetric energy balance. If the former, then storage is zero and Qf is minimal as most Qf is emitted directly into the air volume. If the latter, then advection would seem to be of significance in a heterogeneous urban setting.

This is a statement of conservation across an infinitesimally thin layer of a volume that extends to a depth in the ground where there is no net exchange. For completeness, we add:

**L41**:
  *The surface energy balance for an urban volume can be written (Oke, 1988):*
  $$Q^* + Q_F = Q_H + Q_E + \Delta Q_S + \Delta Q_A \qquad (\text{W m}^{-2}) \qquad (1)$$

  *where $Q^*$ is the net all-wave radiation, $Q_F$ the anthropogenic heat flux, $\Delta Q_S$ the net storage heat flux, $Q_H$ the turbulent sensible and $Q_E$ turbulent latent heat fluxes, $\Delta Q_A$ the net energy transported by advection. These fluxes influence the transfer of heat, mass and momentum and the stability of the urban boundary layer (Oke, 1988).*

2. Section 2.4.3 – Does STEBBS allow for a dynamic setpoint temperature? Most commercial and many residential buildings have setpoints that vary based on management (either BMS or by individual occupants).

Yes, STEBBS allows for setpoint temperatures to be changed at every timestep either to represent specified control schedules and/or human intervention. In the evaluation run, set point temperatures were regulated (between min and max) by relative active occupancy levels (domestic) and building management based on typical percentage of present occupants to workday population. This control approach was applied in relation to the simplification of building representation (i.e. large volume in place of multiple units).

**L302**:

> *The heating of the building fabric modifies the storage heat flux of the urban energy balance (Grimmond et al., 1991; Grimmond and Oke, 1999). Thus this term is tracked and removed from $Q_{F,B}$. Setpoint temperatures are controlled (between minimum and maximum) in relation to occupancy recognising the one-to-many representation of buildings in this model. Domestic instances vary based on the proportion of active occupants to total residential population, whilst non-domestic instances have setpoint temperatures based on occupancy thresholds.*

3. Section 3– are light manufacturing and industrial buildings taken into account in either DASH or GQF? These can be significant energy users in certain areas of larger cities, and might be ignored, potentially explaining part of the underestimation of energy use in the CBD. For that matter, can the authors provide more clarity on how many archetype buildings they use in their modelling? Some prior studies doing similar tasks have suggested 20-30 archetypes are needed to adequately capture citywide variations in residential +commercial energy consumption.

Both DASH and GQF do take light manufacturing and industry into account. DASH increases non-domestic energy use in spatial units with industrial land use informed by energy statistics (i.e. the specific energy demand activity of industry is not captured but magnitude of energy consumption is). GQF includes this with other non-domestic energy consumption at the MSOA resolution.

**L390:**

> *Non-domestic activity varies by workplace appliance types according to the land use (e.g. industrial, office) of the $A_N$ (BEIS 2017a; OpenStreetMap 2017) with appliances (Table D1iii) having greater energy consumption in industrial than commercial areas.*

4. In section 4 (and elsewhere) can authors clarify whether "occupancy" differentiates indoor and outdoor location of individuals? If AC is present, the rejected heat from indoor metabolism includes metabolism + AC energy used to reject it.

Building occupants are assumed to be indoors, but during travel (e.g. walking), or some activities (e.g. outdoor recreation) people occupy outdoor spaces. Heat from metabolism is included as a casual heat source within buildings and would therefore be rejected with AC use. Figure B1 has been updated.

**L217:**

*Both β and M can vary with activity (e.g. office work/sitting, walking, sleeping) and demographics (e.g. age, gender). Occupants are assumed to be indoors when present in an $a_S^N$. When occupants travel and are outside, contributions are made to $Q_{F,M(T)}$.*

**L1170:**

[Figure]

***Figure B1:*** *STEBBS 1-D model simulates building facets/nodes (dots), casual heat sources and heat exchanges. Longwave radiation is absorbed by building facets from the outdoor environment, and shortwave radiation from direct, diffuse and reflected sources.*

5. Lines 631-637 – the errors in estimates of domestic energy consumption seem rather large in general. It would seem that if the building sector archetypes are reasonable, the errors in estimating energy consumption should be much smaller.

Domestic building archetypes are limited, because of data availability, to three domestic types (house, bungalow, flat) and two construction types (by age) for the evaluation. Note there are many occupancy types and building volumes.

6. What are the units in Figure 11 e and f?

Figure 11 updated to show units (W m$^{-2}$).

**L1165:**

[Figure]

***Figure 11 (e, f):*** *Annual average energy consumption at LA scale for (**e**) reference data and (**f**) DASH.*

7. Appendix B – STEBBS: the description of STEBBS suggests that walls and roofs are modelled as single layers with bulk properties. Can the authors clarify whether the conduction equation is solved within these single-layer constructions? If not, they would essentially be assuming steady state conduction through the wall and roof at all times–which would introduce significant errors.

The building components are considered as two layers (indoor and outdoor) with an equal (half) thickness for each layer. In this way the conduction equation is solved across the two layers as well as informing on thermal capacitance. The description in Appendix B has been modified to make this clearer.

**L 734:**

> *STEBBS employs a nodal approach (Foucquier et al., 2013) as found in commonly used simulation tools such as TRNSYS (Klein et al., 2017) and EnergyPlus (Crawley et al., 2000). Each node represents a homogeneous layer within a specified component of the building, with heat transfer equations solved between each node (Figure B1). STEBBS' eight nodes are 2-layers for wall-roof, ground floor and windows; plus a bulk air node and an all internal mass node (calculated as a percentage of total volume). Additionally, there are six nodes associated with the domestic hot water (DHW) system. There are 2-layers for the hot water tank walls and a bulk DHW distribution system, plus a bulk water node for the storage and a distribution node. Effective thermal properties are applied to each component (i.e. a wall cavity and insulation layers are not modelled separately). As this is computationally cheap, it allows multiple instances for each $A_N$ at high temporal resolution. The only latent heat consideration is that of people from metabolic processes (Section 2.4.1).*

Other assumptions in STEBBS may be questionable, as well. I am not familiar with STEBBS. Some more details on this model would be helpful. Additionally, there are standard test cases (from ASHRAE) against which building science models are evaluated and validated. Can the authors present some quantitative data to confirm that the STEBBS model produces accurate estimates of building energy consumption for any specific building (e.g., compare model results with those of a trusted and ASHRAE-validated model such as TRNSYS or EnergyPlus for each of your archetypes). If only~20 archetypes are simulated for a single year, this task is actually rather quick in existing software such as EnergyPlus (or DoE-2) which has already undergone extensive validation and improvement over the past

40+ years. If STEBBS is new (and less validated), can the authors make the case for why they didn't simply go with an existing model?

STEBBS was developed for DASH as a simplified building energy model that is needed to allow for dynamic control of setpoints, occupancy, ventilation, internal loads, etc., as well as consideration of heating and cooling system bulk efficiencies, heat input to the building indoor volume and direct heat rejection to the outdoors. STEBBS does the core calculations for building volume's thermal loading, informed by DASH timestep and parameters.

As the scale of interest is the neighbourhood, STEBBS is not intended to represent individual buildings at the same level of detail/sophistication as done by TRNSYS, EnergyPlus, ESP-r, etc. Given this, a direct comparison between models requires some thought. A comparison has been conducted using EnergyPlus v9.3.0 for the BESTEST Case-600 model setup. This case is chosen as being the most appropriate for the two different modelling approaches. The Case_600.idf file from the EnergyPlus website (originally for v8.1) is modified to run with v9.3.0. The results and discussion have been added to Appendix B.

With respect to the number of effective archetypes: there is of the order of 100,000 different instances run at each 10 min time step. There are six thermophysical properties archetypes. Each of the >25000 neighbourhoods (i.e. OA) have 1–3 domestic instances, and 0–1 non-domestic instances. Within the domestic instances, the occupancy and energy behaviours of households of sizes 1–8 people within the domestic instances are captured.

**L783:**

> *Energy for heating (cooling) is controlled by setpoint temperature with energy added (removed) directly from the indoor air node that is controlled according to a maximum power rating and set system efficiency. The temperature setpoints can change at each timestep allowing both automated and human control to be accounted for. The level of heating (cooling) is further controlled by the difference between indoor air and setpoint temperatures. Internal gains are accounted for as a bulk gain to the indoor air node.*

> *The BESTEST Case 600 single zone building case is used with EnergyPlus (v.9.3.0). to evaluate STEBBS. The EnergyPlus BESTEST model downloaded from the EnergyPlus helpserve website (EnergyPlus, 2020) is modified to run with v9.3.0. Observed London weather data for 2012 (Kotthaus and Grimmond, 2014) are generated using SuPy (Sun and Grimmond, 2019) at an hourly resolution for EnergyPlus and STEBBS. Although EnergyPlus indicates it interpolates sub-hourly weather data for consistency we use both with a 1-hour timestep.*

> *Following EnergyPlus Engineering Reference, the STEBBS external convection coefficient is changed to the DOE-2 method (U.S. Department of Energy, 2020, pg.95-96) for consistency between the models. Note, this is found to have little impact on the results. The internal mass and DHW in STEBBS are reduced in volume to ensure they have negligible impact on results (see Zenodo achive https://doi.org/10.5281/zenodo.3745523 has the BESTEST setup). The bulk building thermal properties in STEBBS are calculated using the BESTEST Case 600 values as presented in ASHRAE 140 (ASHRAE, 2017). Building dimensions for STEBBS are set to give consistent total indoor volume, wall-roof surface area, window area, and floor*

*area. As STEBBS has only one pair of nodes (i.e. 2-layer wall, Figure B1), building geometry and orientation are not represented in STEBBS.*

*The EnergyPlus annual and inter-day heating and cooling dynamics are captured in STEBBS (Figure B2). Both models control the indoor air temperature to within the setpoint limits of 20 (heating) and 27 °C (cooling). EnergyPlus simulates a higher heating and cooling load with more times when the indoor temperature is between (rather than at) the set point temperatures. EnergyPlus also simulates a cooling requirement during the heating season, which STEBBS does not.*

*The modal hourly heating/cooling load differences between the two models are relatively small (Figure B2) but the distribution range is large. The differences are perhaps best attributed to a difference in load control. The EnergyPlus BESTEST case uses the maximum heating (cooling) capacity to add (remove) thermal energy to (from) the building that is likely to result in the observed indoor temperature over shoots, the higher frequency of switching (on-off) for heating and cooling, and need for cooling during heating season as heating and cooling power are set high (100 kW). Whereas to prevent this type of behaviour, STEBBS uses the difference between air and setpoint temperature to help control the heating and cooling power.*

**L1175:**

[Figure]

***Figure B2:*** *BESTEST Case 600 is used with London weather data to evaluate STEBBS relative to EnergyPlus at an hourly time scale for 2012 (a) heating and (b) cooling loads (J), (c) indoor air temperature (d) frequency distribution of hourly differences between EnergyPlus and STEBBS for heating and cooling loads, (e) inter quartile*

*range of hourly differences in winter (Jan, Feb, Mar, Oct, Nov, Dec) and summer (May, Jun, Jul, Aug) loads, and indoor temperatures (whiskers 1% and 99%).*